# Standardized preservation, extraction and quantification techniques for detection of fecal SARS-CoV-2 RNA

Aravind Natarajan [1,2,7], Alvin Han [3,7], Soumaya Zlitni[1,2], Erin F. Brooks [2], Summer E. Vance [2], Marlene Wolfe[4], Upinder Singh[5], Prasanna Jagannathan [3,5], Benjamin A. Pinsky [5,6], Alexandria Boehm [4] & Ami S. Bhatt [1,2 ✉]

Patients with COVID-19 shed SARS-CoV-2 RNA in stool, sometimes well after their respiratory infection has cleared. This may be significant for patient health, epidemiology, and diagnosis. However, methods to preserve stool, and to extract and quantify viral RNA are not standardized. We test the performance of three preservative approaches at yielding detectable SARS-CoV-2 RNA: the OMNIgene-GUT kit, Zymo DNA/RNA shield kit, and the most commonly applied, storage without preservative. We test these in combination with three extraction kits: QIAamp Viral RNA Mini Kit, Zymo Quick-RNA Viral Kit, and MagMAX Viral/Pathogen Kit. We also test the utility of ddPCR and RT-qPCR for the reliable quantification of SARS-CoV-2 RNA from stool. We identify that the Zymo DNA/RNA preservative and the QiaAMP extraction kit yield more detectable RNA than the others, using both ddPCR and RT-qPCR. Taken together, we recommend a comprehensive methodology for preservation, extraction and detection of RNA from SARS-CoV-2 and other coronaviruses in stool.

[1] Department of Genetics, Stanford University, Stanford, CA, USA. [2] Department of Medicine (Hematology, Blood and Marrow Transplantation), Stanford University, Stanford, CA, USA. [3] Department of Microbiology and Immunology, Stanford University, Stanford, CA, USA. [4] Department of Civil and Environmental Engineering, Stanford University, Stanford, CA, USA. [5] Department of Medicine (Infectious Diseases and Geographic Medicine), Stanford University, Stanford, CA, USA. [6] Department of Pathology, Stanford University, Stanford, CA, USA. [7]These authors contributed equally: Aravind Natarajan, Alvin Han. ✉email: asbhatt@stanford.edu

Severe acute respiratory syndrome coronavirus-2 (SARS-CoV-2) is an RNA virus from the *Coronaviridae* family[1] that causes coronavirus disease 2019 (COVID-19). This disease has spread rapidly across the globe and remains a public health threat[2]. COVID-19 is typically considered a respiratory disease, with primary symptoms including cough, sore throat, congestion, anosmia, and dyspnea. However, gastrointestinal (GI) symptoms are also recognized as manifestations of the disease[3,4]. Further, patients shed viral RNA in their stool up to 70 days after disease onset, well after they have cleared the infection from their respiratory tissues[5]. While transmission of SARS-CoV-2 typically occurs through the respiratory tract, some reports indicate the presence of infectious viral particles in patient stool[6,7]. Whether these are truly infectious and have ramifications for public health remains to be definitively demonstrated. However, from an individual patient health perspective, SARS-CoV-2 antigen is found to persist in the GI tract. Further, there is a preliminary hypothesis that persistent SARS-CoV-2 RNA and protein antigen in the GI tract may promote evolution of host humoral immunity to variants of the virus[8]. Relatedly, prolonged viral RNA shedding in stool may indicate a superior immune response[8]. Finally, from an epidemiological perspective, researchers monitor SARS-CoV-2 load in sewage as a proxy for the burden of disease within a population[9]. Taken together, monitoring the fecal shedding of SARS-CoV-2 is vital to fully understanding this pathogen and its effect on patient health in addition to informing public health measures. Therefore, a standardized method to handle and process samples for accurate quantification of viral RNA in stool is critical. Notably, the proposed method should allow for external validity and harmonization of data across studies.

Accurately quantifying fecal shedding of SARS-CoV-2 RNA is challenging. Stool is a heterogeneous sample matrix that contains numerous PCR inhibitors that impede downstream processes like reverse-transcriptase quantitative polymerase chain reaction (RT-qPCR) for quantifying RNA[10]. Further, stool also contains RNases that can rapidly degrade unprotected RNA. Therefore, it is critical that we use appropriate preservatives that protect RNA in stool and employ extraction methods that effectively recover RNA without co-eluting inhibitors or contaminants. In the absence of a comprehensive, standardized protocol, existing studies of SARS-CoV-2 RNA in stool employ methods that have not yet been optimized. Further, the variability in techniques used across studies makes meta-analysis difficult, hindering our overall understanding of the disease. While there is heterogeneity, the majority of existing studies collect and store stool without any preservative[11–13], dilute stool in phosphate-buffered saline (PBS) at the time of RNA extraction, and employ the QIA-Amp Viral RNA Kit from Qiagen for RNA isolation. Unfortunately, the efficiency of these strategies in preserving and extracting SARS-CoV-2 RNA is unknown and has not yet been systematically analyzed. Finally, after RNA extraction, the detection and quantification of RNA by RT-qPCR has elements that have yet to be standardized. While the primer/probe sets used are generally consistent, classifying samples as positive for the presence of SARS-CoV-2 RNA has often been based on arbitrary thresholds set in the absence of a relevant standard curve[14–16]. These experimental inconsistencies and the lack of a clearly validated experimental pipeline contribute significantly to heterogeneity in detection and quantification of viral RNA in stool. To overcome these challenges, we sought to test a variety of accessible and common methods for the preservation, extraction, and detection of viral RNA from stool samples, and present here an optimized pipeline.

In the current study, we present data comparing the performance of three different stool preservatives, three nucleic acid extraction kits, and two PCR-based assays for detecting fecal SARS-CoV-2 RNA. Based on these data, we recommend a pipeline for collecting and processing stool samples for the detection of SARS-CoV-2 RNA. Finally, we validate this standardized pipeline using patient samples collected from a clinical trial. Altogether, our findings here will guide the field toward a more standardized method of robustly measuring the fecal burden of SARS-CoV-2 RNA both in clinical and research settings.

## Results

**Synthetic RNA from ATCC is a reliable positive control and reagent for standard curves.** Accurate quantification of viral loads from standard curves requires reliable positive controls in the form of standardized control RNA at a precisely defined concentration. While many vendors provide synthetic SARS-CoV-2 RNA featuring gene targets recommended by the United States Centers for Disease Control and Prevention (CDC)[17] and the German Centre for Infection Research (DZIF)[18], preliminary studies have revealed that not all of them are at reliable concentrations[19]. Therefore, we tested two synthetic RNA preparations: one from the American Type Culture Collection (ATCC) and one from the United States National Institute of Standards and Technology (NIST) with listed concentrations of $10^5$–$10^6$ and $10^6$ copies/μL, respectively. We chose these positive controls since they are easily accessible to other laboratories and are from reliable sources. A five-point tenfold dilution series from a starting concentration of $10^4$ going down to $10^0$ copies/μL was tested in duplicate droplet digital PCR (ddPCR; Fig. 1a) and quadruplicate RT-qPCR (Fig. 1b) assays targeting the genes for the Envelope protein (E), Nucleocapsid proteins (N1, N2), and RNA-dependent RNA polymerase protein (RdRP)[17,18]. Notably, the NIST standard was provided in two fragments, with fragment 1 bearing the E, N1, and N2 genes and fragment 2 the RdRP gene[20]. The dilution series prepared by two different users working with independent aliquots of the standards revealed ATCC's synthetic RNA standard to be a reliable control with high concordance across reactions targeting the E, N1, and N2 genes (Fig. 1a, b); efficiencies of associated RT-qPCR reactions are 98–101% (Supplementary Data 1). Notably, RdRP proves to be a poor target for the ATCC synthetic SARS-CoV-2 RNA under given reaction conditions, since detection is decreased by an order of magnitude in the ddPCR assay and the RT-qPCR reaction efficiency is compromised (114%). This observation is in keeping with a previous study that found the RdRP primer set to be less sensitive than E, N1, and N2[21]. ddPCR, which allows for absolute quantification, revealed the starting concentration of the ATCC standard to be $10^6$ copies/μL. While the NIST standards also performed with high concordance within replicates across gene targets, the concentration of fragment 2 assayed by targeting RdRP was consistently found to be lower than the stated concentration by two orders of magnitude. Part of this discrepancy may be ascribed to the inefficiency of targeting RdRP as observed previously. Further, one out of a total of eight RT-qPCR reactions assaying the NIST RNA for the E gene at $10^4$ RNA concentration failed to amplify, likely due to an experimental error in the RT-qPCR assay. This result highlights the importance of running RT-qPCR assays in replicates. Given the reliable performance of the synthetic SARS-CoV-2 RNA from ATCC across both ddPCR and RT-qPCR assays testing three target genes, we decided to use this reagent across this study (Supplementary Fig. 2a).

**ddPCR and RT-qPCR assays targeting the N1 gene are reliable means of estimating viral RNA concentration.** We found that primer/probe sets targeting E, N1, and N2 performed comparably in both the ddPCR and RT-qPCR assays based on accuracy of

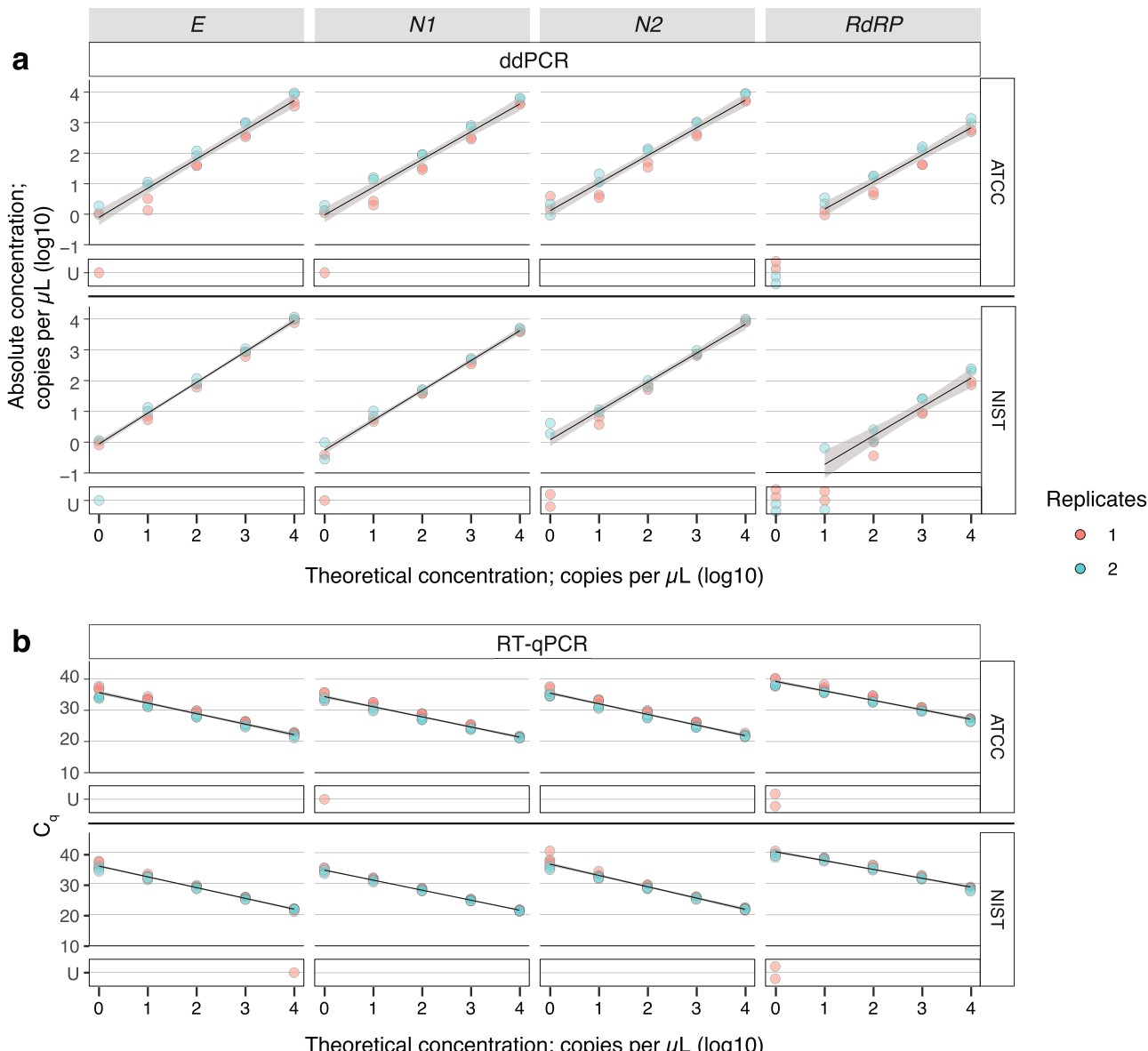

**Fig. 1 Robustness of synthetic SARS-CoV-2 RNA standards from ATCC and NIST.** ddPCR and RT-qPCR assays targeting four SARS-CoV-2 RNA targets (E, N1, N2, and RdRP) across a five-point tenfold concentration range of RNA standards from either ATCC or NIST (indicated on the tab to the right). **a** Theoretical concentrations of RNA are plotted on the x-axis and absolute copy number derived from ddPCR is plotted on the y-axis. All assays were performed in duplicate. **b** Theoretical concentrations of RNA are plotted on the x-axis and $C_q$ derived from RT-qPCR is plotted on the y-axis. All assays were performed in quadruplicate. Replicates in red and blue refer to two independent experiments performed by two users using separate aliquots of samples. Linear regression is plotted in black and 95% confidence interval is shaded in gray. Samples that did not amplify are delineated as U for undetermined and are not included in the linear regression analysis. Associated statistics are summarized in Supplementary Data 1. Source data are provided as a Source data file.

detection with respect to theoretical input concentration and efficiencies of RT-qPCR reactions (Fig. 1a, b and Supplementary Data 1). Among these, we picked N1 as the target gene for the rest of this study (Supplementary Fig. 2a). Given the high degree of concordance across replicate ddPCR and RT-qPCR reactions, we averaged results from replicate reactions in subsequent experiments. Further, since ddPCR allows absolute quantification of viral loads with high sensitivity[22], while RT-qPCR is a more accessible platform for nucleic acid detection, we employed both techniques across the study to be widely informative. In both assays, we used the one-step format that combines the reverse transcription and amplification steps in a single reaction for a simpler protocol.

**Standardized stool samples reveal that preservatives are important and that the Zymo Quick-RNA Viral (ZV) extraction kit performs best.** We tested three different strategies to identify suitable methods of stool preservation for detection of SARS-CoV-2 viral RNA: (a) stool stored without any preservative and resuspended in PBS (PBS), (b) stool preserved in the OMNIgene-GUT tube (OG; DNA Genotek), a commonly used preservation kit in microbiome studies[23], and (c) stool preserved in the Zymo DNA/RNA Shield Collection Kit (ZY; Zymo Research) that is explicitly rated for RNA preservation and virus inactivation.

In parallel, we also tested how these preservation methods interact with three different extraction kits—(a) MagMAX Viral/

Pathogen Nucleic Acid Isolation Kit (MM; Applied Biosystems), a magnetic bead-based protocol that has been successfully used with respiratory samples[24], (b) QiaAMP Viral RNA Mini Kit (QA; Qiagen), a column-based protocol that is used in many studies of fecal SARS-CoV-2 RNA[11–13], and (c) Quick-RNA Viral Kit (ZV; Zymo Research), another column-based protocol that is rated to be compatible with the ZY Stool Collection Kit. All three of these extraction kits are scalable to a high-throughput format and therefore easily adaptable to clinical laboratories and other large-scale efforts.

In order to test and compare all combinations of preservation and extraction methods, we used standardized stool aliquots from NIST. Briefly, these are stool samples collected from a cohort of healthy, omnivorous human donors, which are then homogenized and made available in a tenfold diluted format[25]. We spiked in synthetic SARS-CoV-2 RNA from ATCC (CoV-2 RNA) at two concentrations ($10^3$ and $10^4$ copies/μL of preserved stool sample) in this standardized stool sample and tested the combination of stool preservation and extraction kits to benchmark their performances across multiple target RNA concentrations (Supplementary Fig. 3a). Finally, RNA extractions were performed by two independent users, each in technical duplicates in order to guard against artifacts both across batches by the same user and across users.

Among the stool preservatives, more SARS-CoV-2 RNA was detected in ZY than in OG in both samples spiked with $10^3$ and $10^4$ concentrations of CoV-2 RNA when combined with the MM Extraction Kit (paired $T$ test; $P^{10\wedge3} = 0.003$, $P^{10\wedge4} = 0.002$). We observe the same performance trend with samples spiked with $10^3$ concentration of CoV-2 RNA extracted using QA (paired $T$ test; $P^{10\wedge3} = 0.012$; Fig. 2a and Supplementary Data 5). Notably, ZV outperforms the other extraction kits in samples preserved in OG, and only outperforms QA in those preserved in ZY. Next, we compared the performance of the three extraction kits.

Focusing our attention on the performance of viral extraction kits in combination with the OG and ZY stool preservatives—in OG-preserved samples, ZV outperforms MM by yielding more detectable RNA both in samples spiked with $10^3$ and with $10^4$ concentrations of CoV-2 RNA (Paired $T$ test; $P^{10\wedge3} = 0.011$, $P^{10\wedge4} = 0.012$). Across both stool preservatives, MM and ZV outperform QA (Fig. 2a and Supplementary Data 2). Therefore, in the condition of standardized NIST stool samples spiked with two different concentrations of SARS-CoV-2 RNA we find that the ZY preservative and ZV extraction kit outperform the alternatives.

Notably, in the PBS preservative we detected SARS-CoV-2 RNA at roughly three orders of magnitude lower in eluates extracted from stool spiked with $10^4$ copies/μL of sample compared to OG or ZY. Across stool spiked with $10^3$ copies/μL of sample, we failed to detect any target RNA from PBS-preserved samples. We believe this is because the unpackaged SARS-CoV-2 RNA was degraded by RNases known to be present in stool. While these data suggest that OG and ZY buffers are critical to preserving naked RNA in stool, testing preservatives in the context of unpackaged SARS-CoV-2 RNA may not be representative of clinical samples. This is because we do not yet know whether SARS-CoV-2 RNA shed in stool is in its naked unpackaged state, protected in an encapsulated structure (such as the virus itself, virus-like particles, or host double-membrane vesicles), or a combination thereof.

Hence, we sought to identify a proxy for SARS-CoV-2 that is not known to cause disease in humans and is thus safe to handle in the laboratory at biosafety level 1. We picked bovine coronavirus (BCoV), a virus that belongs to the same genus as SARS-CoV-2, *Betacoronavirus*, in the subgenus *Embecovirus*, sharing this taxonomy with other human pathogens (HCoV-

HKU1 and HCoV-OC43). BCoV and SARS-CoV-2 share a common structural architecture and are both positive stranded RNA viruses. Further, BCoV can be procured as an over-the-counter attenuated vaccine. Prior to stool-based testing, we evaluated the performance of the ddPCR and RT-qPCR assays with the recommended primer/probe set to detect the M gene in BCoV RNA. Toward this, we used RNA extracted directly from the attenuated BCoV vaccine prepared in PBS in the absence of stool. We found both the ddPCR and RT-qPCR assays reliably tracked a seven-point tenfold dilution of the RNA extracts, and the RT-qPCR reaction efficiency of targeting the M gene is 97–100% (Supplementary Fig. 4a, b and Supplementary Data 1). Therefore, we next set out to test the same set of stool preservation and viral RNA extraction methods with the standardized NIST stool samples spiked with BCoV. To assess preservative and extraction kit performance across multiple target concentrations, we spiked BCoV both in its undiluted form and at a tenfold dilution of the stock.

In this experiment, we recovered the target BCoV RNA even in PBS, albeit at lower concentrations compared to other preservation methods (Fig. 2b). Further, the performance of the OG and ZY stool preservatives and the three RNA extraction kits were broadly consistent with previous observations. Briefly, ZY preservative performs better than PBS in all the tested extraction kits, and ZY also performs better than OG when combined with the QA and ZV extraction kits—yielding more detectable target RNA (Fig. 2b and Supplementary Data 2). Given the superior performance of the ZY preservative, we went on to analyze how the three extraction methods fared in this condition. Here ZV surpasses QA at extracting BCoV RNA at both spike-in concentrations. Further, in stool spiked with 1:10 diluted BCoV, ZV also performs better than MM (Supplementary Data 2).

Alongside efforts to extract BCoV RNA from spiked stool samples, each user also extracted RNA directly from the BCoV vaccine without any stool sample. This allows us to evaluate whether the extraction kits interact differently with encapsulated RNA and also serves as a positive control for the extractions. Notably, we find that all extraction kits perform comparably and reliably extract RNA from the BCoV vaccine (Supplementary Fig. 3b).

Finally, we sought to verify our observations using the more commonly used RT-qPCR assay as well. Notably, the RT-qPCR assays (Supplementary Fig. 3c, d) broadly validate trends we observe in the ddPCR assays. All experiments included stool samples with no spiked-in RNA to establish a reliable limit of blank (LoB). RNA extracted from stool samples spiked with BCoV had to be diluted tenfold to arrive at a concentration range accurately quantifiable by ddPCR. Similarly, extracts from BCoV vaccine without stool had to be diluted 100-fold. Finally, given the concordance of results in biological replicates from the same user, we limited the number of replicates to one per user in subsequent experiments.

Taken together, in the NIST omnivore aqueous stool matrix, ZY best preserves both the SARS-CoV-2 naked RNA and encapsulated RNA from BCoV, a SARS-CoV-2-like *Betacoronavirus* (Supplementary Fig. 2b). Further, the ZV extraction kit is also the best performer across both these sample types. Finally, RNA, both in its unpackaged form and when packaged in a virus, is susceptible to loss in PBS without any preservative.

**ZY preservative and QA extraction kits are broadly more effective in non-standardized stool samples.** While the NIST stool samples are a useful, standardized preparation, this processed, pooled, and diluted standardized stool sample is limited in its representation of regular clinical specimens. Therefore, we

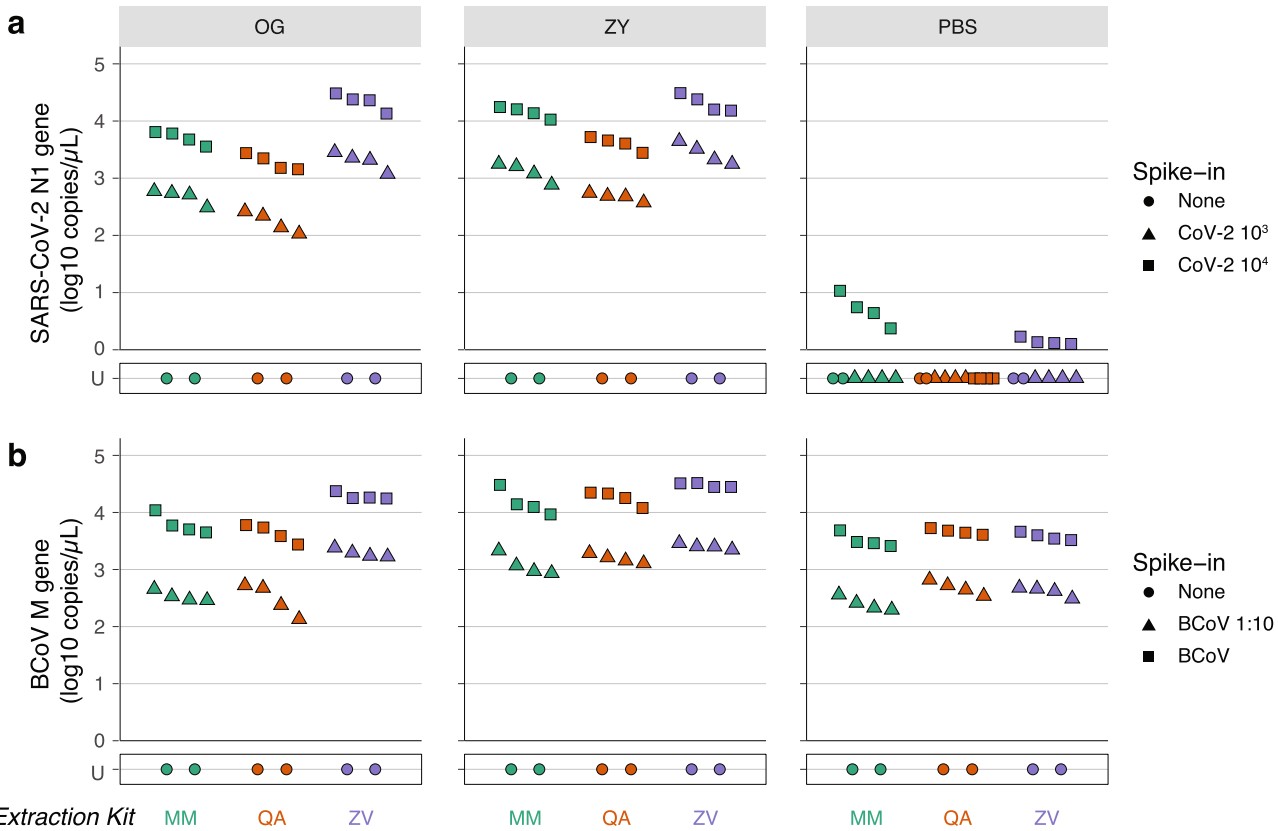

**Fig. 2 Efficacy of preservation and RNA extraction of SARS-CoV-2 and BCoV RNA from standardized NIST stool by ddPCR.** Stool samples collected from omnivorous donors and processed into a single standardized matrix by NIST was spiked with ATCC CoV-2 RNA or BCoV vaccine. Spiked stool was preserved in the OMNIgene-GUT Kit (OG), Zymo DNA/RNA shield buffer (ZY), and PBS (as indicated in the tab on the top). RNA was extracted from these samples by two independent users, each in duplicate, using the MagMAX Viral/Pathogen Kit (MM; green), QIAamp Viral RNA Mini Kit (QA; orange), or Zymo Quick-RNA Viral Kit (ZY; purple) as indicated on the x-axis. RNA was assayed using ddPCR. **a** Absolute concentration of SARS-CoV-2 RNA assayed by ddPCR targeting the N1 gene is plotted on the y-axis. NIST stool matrix was spiked with $10^3$ (triangle) or $10^4$ (square) copies of ATCC synthetic SARS-CoV-2 RNA. **b** Absolute concentration of BCoV RNA assayed by ddPCR targeting the M gene is plotted on the y-axis. NIST stool matrix was spiked with 1:10 diluted (triangle) or undiluted (square) BCoV vaccine. Control samples with no spiked in RNA (none; circle) were included in duplicate to estimate LoB. U stands for undetermined and marks samples with no detectable RNA above LoB. Two-sided paired $T$ tests were performed on $n = 4$ independent extractions for each spike-in condition. Associated statistics are summarized in Supplementary Data 2. Source data are provided as a Source data file.

next tested the combinations of preservatives and viral RNA extraction kits using undiluted and unprocessed stool samples from healthy donors, spiked with the SARS-CoV-2 RNA and BCoV standards. We picked the lower concentrations of both the SARS-CoV-2 RNA ($10^3$) and BCoV (1:10 dilution) from our previous analysis to challenge the sensitivity of the combinations of preservation, extraction, and detection techniques tested here.

We acquired stool samples from two healthy stool donors, one on an omnivorous diet (Omni) and the other on a vegetarian diet (Veg) (Supplementary Fig. 5a). Across conditions, the concentrations of target RNA detected from these matrices were lower than those from the NIST samples by around an order of magnitude (Figs. 2a and 3a). We used ddPCR to assay the performance of the preservatives and observed that, in samples spiked with SARS-CoV-2 RNA, OG and ZY perform comparably and better than PBS when paired with either the QA of ZV extraction kits. However, ZY performs better than both OG and PBS when combined with the MM extraction kit (Fig. 3a and Supplementary Data 3). In the best performing preservative, ZY, all extraction kits perform comparably. Notably, PBS continues to perform poorly, yielding no detectable target RNA in all but one extraction. These results based on unprocessed non-standardized stool samples suggest that it is best to preserve

samples in the ZY buffer and that, in this preservative, all three extraction kits can be used with comparable results.

In the case of RNA encapsulated in BCoV, the two preservatives, OG and ZY, perform comparably unless combined with the QA extraction kit where ZY outperforms OG (paired $T$ test; $P = 0.017$; Fig. 3b and Supplementary Data 3). Next, focusing on samples preserved in ZY, all three extraction kits yield comparable amounts of detectable viral RNA. Further, control extractions included in this batch of assays, with only the BCoV vaccine without any stool, also yielded comparable amounts of RNA across kits (Supplementary Fig. 5b). We note that RNA extracted from stool samples spiked with BCoV had to be diluted ten fold to arrive at a concentration range accurately quantifiable by ddPCR and those from BCoV vaccine without stool had to be diluted 100-fold.

Unlike the previous experiment with standardized diluted NIST stool, in this set of samples based on unprocessed healthy stool, we observe differences in the performance of ddPCR (Fig. 3a, b) and RT-qPCR assays (Supplementary Fig. 5c, d). Interestingly, we detected BCoV from the PBS sample extracted with the ZV kit in the RT-qPCR assay, albeit at a high $C_q$ value (Supplementary Fig. 5d), but not in the ddPCR assay (Fig. 3b). This is one exception among all the assays performed in this

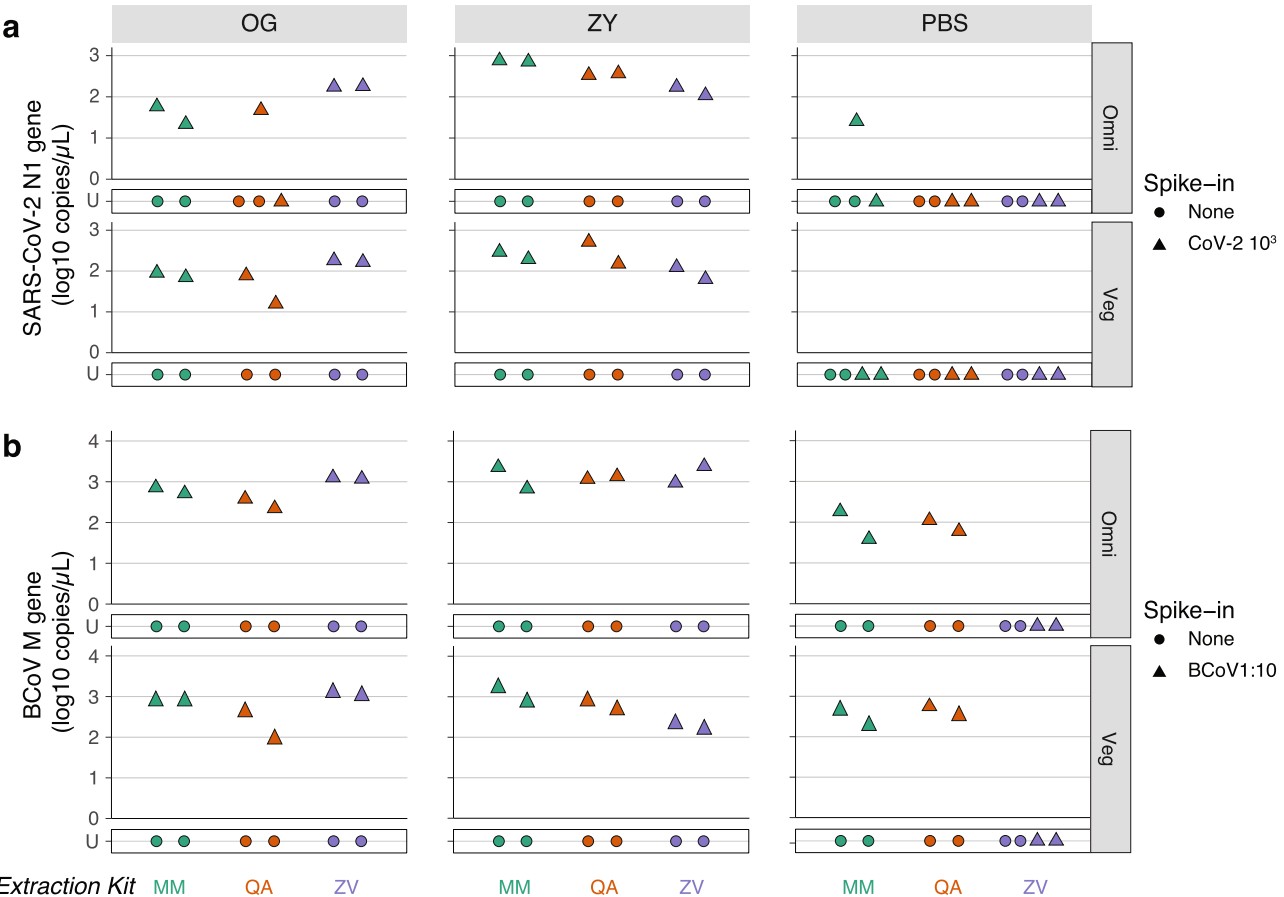

**Fig. 3 Evaluating preservation and extraction of SARS-CoV-2 and BCoV RNA from non-standardized stool samples using ddPCR.** Stool samples were collected from healthy omnivorous (Omni) and vegetarian (Veg) donors and spiked with ATCC CoV-2 RNA or BCoV vaccine. Spiked stool was preserved in the OMNIgene-GUT Kit (OG), Zymo DNA/RNA shield buffer (ZY), and PBS (as indicated in the tab on the top). RNA was extracted from these samples by two independent users using the MagMAX Viral/Pathogen Kit (MM; green), QIAamp Viral RNA Mini Kit (QA; orange), or Zymo Quick-RNA Viral Kit (ZY; purple) as indicated on the x-axis. RNA was assayed using ddPCR. **a** Absolute concentration of SARS-CoV-2 RNA assayed by ddPCR targeting the N1 gene is plotted on the y-axis. Healthy stool samples were spiked with $10^3$ (triangle) copies of ATCC synthetic SARS-CoV-2 RNA. **b** Absolute concentration of BCoV RNA assayed by ddPCR targeting the M gene is plotted on the y-axis. Healthy stool samples were spiked with 1:10 diluted (triangle) BCoV vaccine. Control samples with no spiked in RNA (none; circle) were included in duplicate to estimate LoB. U stands for undetermined and marks samples with no detectable RNA above LoB. Two-sided paired $T$ tests were performed on $n = 4$ independent extractions for each target. Associated statistics are summarized in Supplementary Data 3. Source data are provided as a Source data file.

study and likely a false positive. Next, by and large, we were unable to detect the BCoV target in RNA extracted using the MM kit when assayed by RT-qPCR (Supplementary Fig. 5d), even though ddPCR enables detection (Fig. 3b). This observation makes us suspect that PCR inhibitors are being coeluted with RNA when using the MM kit; RT-qPCR is susceptible to PCR inhibitors, while ddPCR is much less affected. Given these observations, we conclude that QA performed most reliably at yielding detectable RNA from BCoV spiked into non-standardized stool.

Overall, these experiments comparing the performance of preservatives and extraction kits on non-standardized stool samples revealed that ZY yields more detectable target RNA than OG and PBS (Supplementary Fig. 2b). Further, we are still unable to detect RNA from samples stored in PBS when trying to recover the unpackaged ATCC synthetic RNA spiked into stool. Finally, while all extraction kits perform comparably at extracting unpackaged RNA, QA performs more reliably than MM and ZY at extracting BCoV-encapsulated RNA.

**ZY collection and preservation is more effective than OG in the real-world setting.** Experiments so far studied defined stool

samples spiked with a known amount of target RNA and transferred to collection kits in a precise, controlled laboratory environment. This is useful for testing kits head to head. However, in reality, stool samples are likely to be collected by patients or healthcare practitioners outside of well-controlled laboratory spaces. Additionally, transit of the viral RNA through the GI tract may affect its detection in a manner not captured by the spiked-in stool samples. Therefore, it is important to compare the performance of the OG and ZY stool preservatives in this practical use-case setting with samples from COVID-19 patients.

To this end, we leveraged an ongoing large-scale study that captures the dynamics of fecal SARS-CoV-2 viral RNA shedding. Briefly, this study collected stool samples from COVID-19 outpatients who were enrolled in a clinical trial of Peginterferon Lambda-1a[26] in both the OG and ZY preservatives. RNA was extracted from these samples using QA and assayed to determine viral load using RT-qPCR.

From this data set, we picked instances of paired OG and ZY viral loads determined from samples collected from the same patient at the same time. Out of 240 such samples from 98 independent COVID-19 outpatients, 122 stool samples did not yield a detectable amount of target RNA in either preservative

and were left out of further analysis. Taking the 118 paired samples from which we detected the viral RNA targets in at least one of the preservatives, we plotted their $\log_{10}$-transformed concentrations in a scatter plot (Fig. 4). Here we fitted a linear regression, excluding samples that yielded RNA in only one of the two preservatives since these skewed the regression. Notably, 27 of these paired samples yielded detectable RNA only in ZY, in comparison to 13 in only OG. The linear regression from the paired samples stored in OG and ZY reveals that, among samples for which both samples tested positive, OG samples had a roughly 60% lower detected concentration of RNA. Finally, we also calculated the mean of the differences between the $\log_{10}$-transformed viral RNA concentrations from these paired samples, including ones that were only detected in one of the two preservatives. This revealed that ZY-preserved samples yielded more RNA than OG samples ($1.202 \pm 0.939$ $\log_{10}$ copies per µL in ZY versus $0.821 \pm 0.759$ $\log_{10}$ copies per µL in OG) by $0.381$ $\log_{10}$ units (or ~2.4 times more) (two-sided paired $T$ test; $t = -5.103$, df $= 117$, $P < 0.001$).

While the ZY preservative may be more effective at protecting RNA, it is also possible that the ZY collection kit ends up with more stool compared to the OG kit. In order to address this question, we estimated how much of the sample from either of these kits is actually composed of stool. To this end, we randomly selected paired samples collected in the OG and ZY tubes from the biobank of stool samples collected from COVID-19 out-patients enrolled in the aforementioned clinical trial of Peginter-feron Lambda-1a. Specifically, each of these pairs was collected from the same patient at the time of enrollment in the study. We took two biopsy punches from each of these ten stool samples and measured their wet weight. Next, we dried these samples on a heat block for 72 h and measured their dry weight. The percentage of dry weight to wet weight represents the proportion of patient stool biomass in the original sample. We found that $31.4 \pm 1.6\%$ of sample weight in the ZY preservative corresponds to stool biomass, compared to $13.6 \pm 3.3\%$ of sample weight in the OG preservative (Table 1; paired $T$ test; $P = 5.49E-6$). This roughly threefold difference in stool biomass tracks closely with the threefold difference we observe above in the performance of the two kits. Therefore, likely, the two kits preserve and yield comparable amounts of detectable SARS-CoV-2 RNA, when accounting for the amount of input stool.

However, the difference in stool biomass across the two kits is surprising to us, since reading the manufacturer's instructions suggests that the OG kit would end up with a marginally higher concentration of stool. In fact, the experiments with stool from NIST and healthy donors described in this work followed these instructions and added 500 mg of stool to OG (containing 2 mL of buffer) and 1000 mg of stool to ZY (containing 9 mL of buffer). We suspect that this difference in stool input we observe in the clinical samples may be the effect of the format of the two kits. Specifically, the OG kit is composed of a specific receptacle of defined volume to collect stool, while the ZY kit is just a standard collection tube with a proprietary buffer (Supplementary Fig. 6). The ZY kit has plenty of room in the tube above the buffer level, so study subjects may have been inclined to load more stool in the ZY kit.

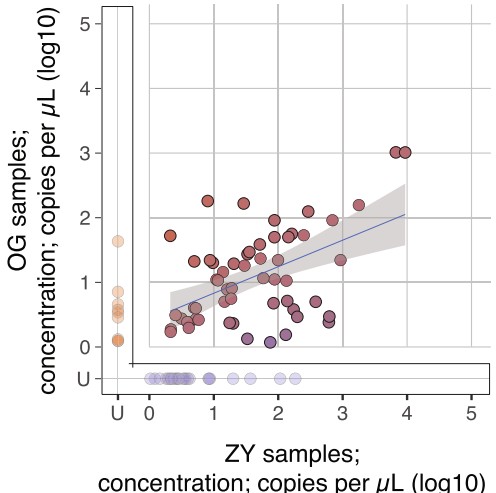

**Fig. 4 Relationship between yields of SARS-CoV-2 RNA extracted from clinical samples stored in two different preservatives.** Paired stool samples were collected in the OMNIgene-GUT Kit (OG) and Zymo DNA/RNA shield buffer (ZY) preservatives from COVID-19 outpatients enrolled in a clinical trial of Peginterferon Lambda-1a. RNA from these samples had been extracted using the QIAamp Viral RNA Mini Kit (QA), assayed by RT-qPCR targeting the N1 gene and previously reported. Presented is a scatter plot of the reported concentrations of paired stool samples with concentrations derived from the ZY-preserved samples on the *x*-axis and from the OG-preserved samples on the *y*-axis. A linear regression is plotted in black and 95% confidence interval is shaded in gray, centered around the line of best fit. Samples that are reported to not have amplified are delineated as U for undetermined and are not included in the linear regression analysis. This breaks down as 27 samples that were detected only in the ZY preservative, 13 that were detected only in the OG preservative, and 78 that were detected in both. The associated regression equation generated from $n = 78$ extractions in which both ZY and OG yielded detectable RNA is $y = 0.531 + 0.381x$, $R^2_{adj} = 0.203$ and $p = <0.001$ ($F$-statistic $= 20.55$ on df $= 1$ and 76, two-sided test). The color gradient from purple to orange represents the ratio of the concentration of RNA derived from the ZY-preserved sample (purple) to that derived from the OG-preserved sample (orange). Source data are provided as a Source data file.

**Table 1 Measurement of wet weight and dry weight from five paired stool samples collected from COVID-19 patients in OG and ZY.**

| Sample ID | ZY Wet weight (grams) | ZY Dry weight (grams) | ZY Percent stool (%) | OG Wet weight (grams) | OG Dry weight (grams) | OG Percent stool (%) |
|---|---|---|---|---|---|---|
| Lambda_248 | 0.119 | 0.036 | 30.252 | 0.172 | 0.031 | 18.023 |
| Lambda_327 | 0.145 | 0.043 | 29.655 | 0.117 | 0.011 | 9.402 |
| Lambda_223 | 0.101 | 0.034 | 33.663 | 0.168 | 0.023 | 13.690 |
| Lambda_264 | 0.096 | 0.031 | 32.292 | 0.117 | 0.018 | 15.385 |
| Lambda_292 | 0.157 | 0.049 | 31.210 | 0.111 | 0.013 | 11.712 |
| Average | | | 31.414 | | | 13.642 |
| Std. deviation | | | 1.606 | | | 3.314 |

Taken together, we find that the ZY kit yields more detectable RNA than the OG kit both with samples prepared in strictly controlled experimental conditions carried out in the laboratory and in those collected in the field by patients (Supplementary Fig. 2c). This superior performance may be the result of a better preservative, differential usage of these kits, or a combination thereof.

**Comparing the performance of extraction kits on clinical samples collected in the ZY preservative**. Given the superior performance of the ZY preservative in both standardized and clinical samples, we next tested how the three extraction kits perform with real-life clinical samples preserved in this modality.

In order to test the extraction kits, we picked 20 random samples from a biobank of stool collected as part of the aforementioned Peginterferon Lambda-1a clinical trial[26]. We picked samples that were preserved in ZY on the day of enrollment, when we expect a higher fecal viral load. Recognizing that there is a high cost to false negatives in the detection of SARS-CoV-2 viral RNA in stool samples, we sought to incorporate a reliable control to track efficiency of RNA extraction, without compromising the yield of the target SARS-CoV-2 RNA. Hence, we took two aliquots of the stool from each of the chosen 20 clinical samples and spiked undiluted BCoV in one of them as a reference extraction control (Supplementary Fig. 7a). We then extracted RNA from the 40 stool samples using the three extraction kits resulting in 120 RNA samples.

At the outset, while we successfully extracted and detected BCoV in all spiked samples, we did not detect any SARS-CoV-2 RNA in stool collected from nine patients across any extraction kit. Therefore, we excluded these samples from further analysis. We next looked at samples spiked with BCoV that yielded detectable SARS-CoV-2 RNA in at least one extraction kit. Among these, we find that QA outperforms ZV ($P = 0.014$), yielding more detectable target RNA. However, other pair-wise comparisons of SARS-CoV-2 RNA yields across kits are comparable (Supplementary Fig. 2c and Supplementary Data 4). Comparing the yield of SARS-CoV-2 RNA in samples with (Fig. 5a) and without (Supplementary Fig. 7b) spiked in BCoV reveals that the addition of this control does not significantly affect the yield of RNA (Fig. 5b) with the exception of sample # 10 extracted using MM. In this case, where we expected consistency across samples with and without the BCoV spike in, inexplicably, we find that the stool sample with spiked BCoV does not yield SARS-CoV-2 RNA while the unspiked sample does. We reran the ddPCR assay and observed the same result. Notably, the BCoV spiked sample that does not yield detectable SARS-CoV-2 RNA yields BCoV RNA, revealing that the extraction and detection steps are successful. Therefore, we suspect that this anomaly observed in 1 out of the 20 samples tested to be the result of experimental error in sample preparation.

In a subset of these samples ($n = 5$), we also report the detected copies of N1 gene per gram of stool to normalize the copies of SARS-CoV-2 RNA to the amount of stool placed into the preservative by patients. This does not alter our conclusions regarding the best extraction kit, but given the differing input of stool in various preservative options, we believe that reporting detected copies per gram of stool where possible will best harmonize reported viral loads of SARS-CoV-2 in feces (Supplementary Note 1).

We next analyzed the cumulative yield of BCoV RNA from each of these clinical samples and found that samples extracted with the MM and QA kits performed comparably and reliably, whereas BCoV RNA detection was most variable in samples extracted with the ZV kit (Fig. 5c). Therefore, we demonstrate

here how an easily accessible, over-the-counter attenuated BCoV vaccine can be leveraged as a reliable spike-in control.

All results considered, we recommend using the ZY preservative to collect stool samples and the QA extraction method to purify SARS-CoV-2 RNA (Supplementary Fig. 8). In instances where variability in extracted RNA yield or coelution of polymerase inhibitors are anticipated, we suggest spiking in 10 μL of BCoV vaccine to 500 μL of stool prior to storage and extraction in order to guard against false negatives. We have validated here that BCoV serves as a reliable control and does not affect the detection of SARS-CoV-2 RNA.

## Discussion
Fecal shedding of SARS-CoV-2 RNA is emerging as a key manifestation of COVID-19 infection with vast implications for patient health and in the epidemiology of the disease. However, methods to collect and preserve patient samples and to extract viral RNA for the robust detection and quantification of SARS-CoV-2 remain underexplored. Therefore, we compare strategies for each of these steps in the testing of fecal samples and report here an optimized methodology. We have focused our efforts on reagents that are easily available and kits that are scalable to a high-throughput format, therefore enabling straightforward adoption in research and clinical laboratories.

We tested three different strategies for sample collection and preservation. First, the most common strategy involves collecting stool without any preservative[27–35]. These samples are resuspended in PBS for viral RNA extraction[11]. Next, we also tested the OG preservative that is widely used in stool collection for gut microbiome analysis[23]. Finally, we included ZY as a sample preservation method that is explicitly marketed for RNA preservation. Across three different types of stool samples, ZY consistently performed better than OG and PBS, enabling both the recovery of naked, unpackaged SARS-CoV-2 RNA and BCoV RNA encapsulated in a *Betacoronavirus* similar to SARS-CoV-2. Most importantly, analysis of data from a large study of outpatients with mild-to-moderate COVID-19 further validated the conclusion that ZY was the most effective preservation method. Conclusions from our study in combination with existing evidence that SARS-CoV-2 RNA is susceptible to degradation from freezing stool samples without any preservative[12] highlights the importance of storing stool samples in an appropriate buffer.

Next, we compared three different extraction kits for their potential to effectively isolate viral RNA. Two of these kits, QA and ZY, are column-based kits, while MM is based on magnetic beads. We tested these kits by performing replicate nucleic acid extractions of stool samples prepared in the laboratory spiked with SARS-CoV-2 synthetic RNA or BCoV vaccine. Here we found that the performance of the extraction kits was influenced by the preservative, nature of stool, and the target RNA. We focus our discussion on the performance of the extraction kits in combination with the best performing preservative, ZY. Here we observe that ZV most effectively extracted both the unpackaged SARS-CoV-2 RNA and the packaged BCoV RNA from the standardized, diluted NIST stool samples. However, from non-standardized healthy stool samples and clinical samples, QA performed more consistently, yielding detectable viral RNA across conditions. Notably, while MM performed well in many of the experiments, we find preliminary evidence that this protocol may allow the co-purification of PCR inhibitors. We glean this observation from experiments performed with BCoV spiked into non-standardized healthy stool samples. Taken together, we recommend using the QA extraction kit in tandem with the ZY preservative as a strategy for the robust and sensitive detection of SARS-CoV-2 RNA from stool (Supplementary Fig. 8).

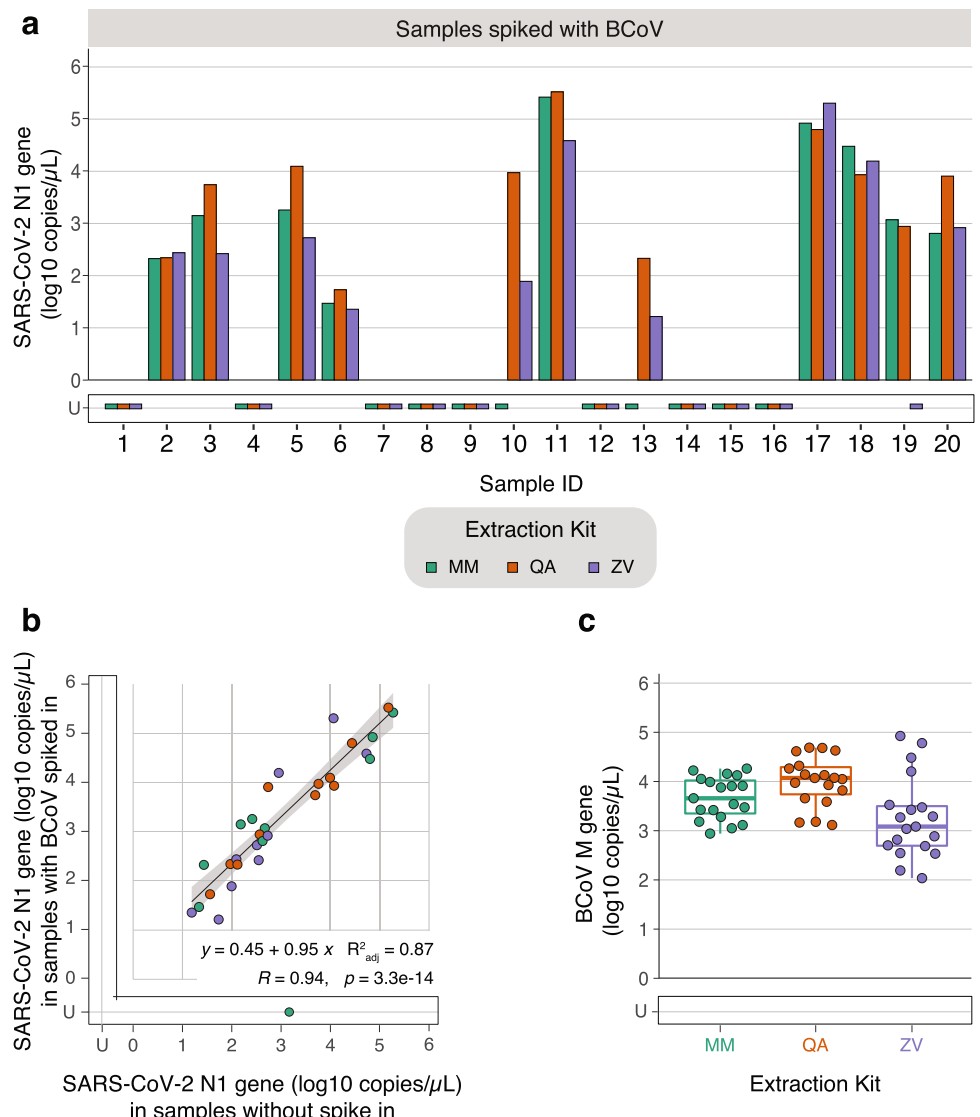

**Fig. 5 Testing efficiency of three extraction kits using clinical samples stored in the ZY preservative and spiked with BCoV.** Stool samples were collected in the Zymo DNA/RNA shield buffer (ZY) preservative from 20 COVID-19 outpatients enrolled in a clinical trial of Peginterferon Lambda-1a. All samples were spiked with 10 μL of undiluted BCoV vaccine. In parallel, the same set of samples were processed without any spike-in. RNA from these samples were extracted using the MagMAX Viral/Pathogen Kit (MM; green), QIAamp Viral RNA Mini Kit (QA; orange), or Zymo Quick-RNA Viral Kit (ZY; purple). U stands for undetermined and indicates samples without no detectable RNA above the LoB. **a** RNA from samples with BCoV spiked in were assayed for SARS-CoV-2 RNA using ddPCR targeting the N1 gene. Anonymized sample identities are listed on the *x*-axis and absolute concentration is listed on the *y*-axis. Two-sided paired *T* tests were performed on $n = 11$ independent extractions (9 samples for which no RNA was detected across any kit were excluded). **b** Scatter plot of the absolute concentration of SARS-CoV-2 RNA derived from samples without any spike-in (*x*-axis) versus those with 10 μL of undiluted BCoV spiked in (*y*-axis), measured using ddPCR targeting the N1 gene. Thirty samples that yielded no detectable RNA above LoB in both sets of samples were left out of the plot. A linear regression is plotted in black and 95% confidence interval is shaded in gray, centered around the line of best fit. Samples with undetermined concentrations in either one of the cases are not included in the linear regression analysis. The associated regression equation generated from the $n = 29$ remaining extractions is $y = 0.450 + 0.950x$, $R^2_{adj} = 0.870$, $R = 0.940$, and $p = 3.3E-14$. **c** RNA extracted from samples with BCoV spiked were assayed for BCoV RNA using ddPCR targeting the M gene. A cumulative box plot shows the absolute concentrations of BCoV RNA across the clinical samples ($n = 20$ independent biological samples). The lower bound of the box marks the first quartile, the higher bound of the box marks the third quartile, the horizontal line marks the median, and the whiskers mark the outlier-removed minima and maxima. Source data are provided as a Source data file.

Stool-based testing of SARS-CoV-2 offers unique applications in healthcare. With emerging evidence that prolonged shedding of SARS-CoV-2 RNA in stool may be linked to an improved immune response[8], there may be an opportunity to leverage fecal testing of RNA as a prognostic marker. Further, if the limited evidence of possible oral-fecal transmission of SARS-CoV-2 proves true, our ability to reliably test stool samples would be vital to controlling the spread of the pandemic as well as to inform

healthcare practices, such as fecal microbiota transplants. Finally, this option protects healthcare practitioners from having to be in close proximity to patients during sample collection. In all of these applications, it is critical to incorporate strategies to mitigate false negatives. Such false negatives may arise from errors in sample preservation, RNA extraction, and presence of inhibitors that affect detection through PCR-based methods. Therefore, in this study, we also evaluate potential controls to guard against

instances of such false negatives. We find that the widely accessible, safe BCoV vaccine can be effectively spiked into stool samples prior to storage and extraction. Recovery of BCoV RNA assayed by targeting the M gene serves as a reliable metric of variation across batches of RNA preparations without affecting the yield of SARS-CoV-2 RNA in the samples. We believe BCoV to be a valuable proxy for SARS-CoV-2 since they belong to the same genus, *Betacoronavirus*, and predominantly share viral architecture. Therefore, using BCoV as a spiked-in control will help gain confidence in negatives as true negatives rather than a result of experimental artifacts (Supplementary Fig. 8).

Further, given the clinical implications, it is equally important to avoid false positives in the detection of SARS-CoV-2 RNA in stool. To this end, it is vital to establish a LoB with every batch of experiments. This allows the confident identification of true positive samples over experimental noise. Guidelines from the Clinical and Laboratory Standards Institute (EP17-A) provide a roadmap for a rigorous evaluation of LoB[36]. We recognize that this is a high bar for non-clinical research laboratories to meet. Alternatively, as demonstrated here, including comparable control stool samples from NIST or healthy donors in every batch of viral RNA extraction and detection will also serve to boost confidence in the detection of SARS-CoV-2 RNA as being a true positive (Supplementary Fig. 8).

Finally, it is important to be able to quantify the viral RNA load in stool as a potential indicator of the state and prognosis of infection in patients. To this end, while ddPCR provides a powerful platform capable of determining the absolute concentration of RNA, we recognize that this may be cost prohibitive and inaccessible. Therefore, additionally, we demonstrate here experimental strategies that enable the adoption of the more accessible RT-qPCR assay to enable the accurate detection and relative quantification of viral load in samples (Supplementary Fig. 8). Lastly, given differing amounts of stool collected by every patient and in different experiments, we recommend reporting quantified viral RNA load in terms of copies per gram of stool. This enables a normalized dataset that will allow us to harmonize reported fecal viral loads of SARS-CoV-2 RNA across studies.

SARS-CoV-2 has been a deadly pathogen causing extensive morbidity and mortality. Given the current understanding of coronaviruses, it is likely that SARS-CoV-2 will not be the last virus of this nature to cause an epidemic. Further, many coronaviruses are capable of infecting the GI tract. In this context, we hope that the current work helps create a roadmap for fecal testing of coronavirus infections enabling the robust detection and quantification of viral RNA in stool.

## Methods

### Preparation of stool samples spiked with SARS-CoV-2 RNA or BCoV-attenuated virus.
We used two types of non-clinical stool samples in this study. The first set of samples were acquired from the United States NIST[25] and stored at −80 °C. The second were acquired from healthy donors collected with informed consent as part of the Stanford Institutional Review Board (IRB) protocol #42043 (PI: Ami S Bhatt; Title: Genomic, Transcriptomic and Microbiological Characterization of Human Body Fluid Specimens) and stored at −80 °C without any preservatives. Finally, samples and data from patients were collected as per Stanford IRB protocol #55619 (PIs: Upinder Singh, Prasanna Jagannathan; Title: A Phase 2 Randomized, Open Label Study of a Single Dose of Peginterferon lambda-1a (Lambda) Compared with Standard Supportive Care in Outpatients with Mild COVID-19).

In most studies, stool samples are collected and stored without a preservative[11–13]. They are then resuspended in PBS in a 1:5 ratio (w:v) prior to RNA extraction. As a proxy for these samples, we added 1000 mg of stool to 5 mL of PBS and describe these as PBS-preserved samples throughout this study. Separately, we also stored samples in the OG (DNA Genotek; Catalog # OMR-200) and ZY (Zymo Research; Catalog # R1100-250) according to the manufacturers' instructions. Briefly, we added 500 mg of stool to the OG tube containing 2 mL of preservative to prepare the OG samples and 1000 mg stool to the ZY kit with 9 mL of buffer to prepare the ZY samples. Given shortages in the supply of the ZY kit, we also resorted to recreating this kit in house using 9 mL of the DNA/RNA Shield

buffer (Zymo Research; Catalog # R1100-1L; Lot # ZRC195881) in a 15 mL centrifuge tube (VWR; Catalog # 89039-666). Subsequently, where listed, we spiked in synthetic SARS-CoV-2 RNA from ATCC (Catalog # VR-3276SD, Lot # 70034237) at a final concentration of either $10^3$ or $10^4$ copies/μL of preserved stool sample. For samples spiked with the control BCoV, we prepared BCoV by resuspending one vial of lyophilized Zoetis Calf-Guard Bovine Rotavirus-Coronavirus Vaccine (Catalog # VLN 190/PCN 1931.20) in 3 mL of PBS to create an undiluted reagent as per the manufacturer's instructions. We then added 60 μL of either undiluted or diluted BCoV (1:10 in PBS) to 3 mL of preserved stool sample. To create an extraction blank control, an equivalent volume of PBS was added to samples labeled 'None'. All samples were then stored in 1.5 mL DNA LoBind tubes (Fisher Scientific; Catalog # 13-698-791) and immediately frozen at −80 °C.

### Preparation of clinical stool samples.
Clinical samples were collected and stored from patients participating in an interventional study of Peginterferon Lambda-1a[26]. Briefly, study subjects were requested to collect samples in both the OG and ZY tubes according to the manufacturer's instructions, and samples were stored at room temperature for up to 7 days before being processed into cryovials and frozen at −80 °C until subsequent use. We directly used OG- and ZY-preserved samples in the subsequent extraction steps. Where mentioned, samples were spiked with 10 μL of attenuated BCoV vaccine per 500 μL of preserved stool sample after thawing an aliquot for extraction.

### Viral RNA extraction.
We spun down 600 μL of each preserved stool sample at $10,000 \times g$ for 2 min to remove solids from the sample. We then processed 200 μL of supernatant according to the manufacturer's instructions for QA (Qiagen; Catalog # 52906, Lot #166024216) and ZY (Zymo; Catalog # R1035, Lot #206187). For supernatant processed using MM (Thermo Fisher Scientific; Catalog # A42352, Lot # 2009063, 2008058), we followed the manufacturer's instructions with the following exception: samples were processed in 1.5 mL DNA LoBind tubes rather than 1.5 mL deep-well plates. We eluted RNA from each sample in 60 μL of the elution buffer included in each kit. The eluted RNA was stored in a 96-well plate at −80 °C.

### Quantification of viral RNA by RT-qPCR.
We assembled the RT-qPCR reaction using a Biomek FX liquid handler by adding 5 μL of eluted RNA to 5 μL of TaqPath 1-Step RT-qPCR CG mastermix (Applied Biosystems, Catalog # A15300, Lot 2293196), 8.5 μL of nuclease-free water (Ambion, Catalog # AM9937, Lot 2009117), and 1.5 μL of primer/probe mix. The primer/probe mix was composed of 200 nM each of forward primer, reverse primer, and probe (Elim Biopharmaceuticals) with sequences summarized in Supplementary Data 5. We designed the probes with 5′ fluorescein and 3′ 5-carboxytetramethylrhodamine dyes.

Our RT-qPCR analysis is guided by the Minimum Information for Publication of Quantitative Real-Time PCR Experiments (MIQE) guidelines[37], and the recommended associated checklist is included in Supplementary Data 6. We used the QuantStudio 12K Flex (Applied Biosystems) to amplify the template using the following thermocycling program: 25 °C for 2 min, 50 °C for 15 min, 95 °C for 2 min, 45 cycles of 95 °C for 15 s and 55 °C for 30 s with ramp speed of 1.6 °C/s at each step. We calculated the quantification cycle ($C_q$) value using the Design and Analysis software (Version 2.4.3; Thermo Fisher Scientific). Standard curves for quantification were generated using a five-point tenfold dilution of the SARS-CoV-2 ATCC standard from $10^4$ to $10^0$ copies/μL of template. We calculated the concentration of RNA using a linear regression of the standard curve. We established LoB on a plate-by-plate basis; specifically, we turned to the specific plate that an experimental sample was assayed on and picked the lowest $C_q$ among the following controls run in the same plate: the y-intercept of the line of best fit from the standard curve, none (no RNA or BCoV spiked) stool samples, water, and elution buffers from the RNA extraction kits as listed in the relevant experiments. RNA concentrations from reactions with $C_q$ values below the LoB were defined as Undetermined. The concentration of RNA from technical duplicate RT-qPCR reactions were averaged. If one of the two technical duplicate reactions failed to amplify within the range of the standard curve, the viral concentration from that sample was treated as Undetermined.

### Quantification of viral RNA by ddPCR.
Our ddPCR analysis is guided by the Droplet Digital PCR Applications Guide on QX200 machines (BioRad)[38] and the digital MIQE guidelines (recommended checklist is included in Supplementary Data 7)[39]. We assembled the ddPCR reaction using a Biomek FX liquid handler by adding 5.5 μL of eluted RNA to 5.5 μL Supermix, 2.2 μL reverse transcriptase, 1.1 μL of 300 nM dithiothreitol (DTT), 1.1 μL of 20× Custom ddPCR Assay Primer/Probe Mix (BioRad, Catalog # 10031277), and 6.6 μL of nuclease-free water (Ambion, Catalog # AM9937, Lot 2009117). The Supermix, reverse transcriptase, and DTT are from the One-Step RT-ddPCR Advanced Kit for Probes (BioRad, Catalog # 1864021). We then processed the assembled reactions on a QX200 AutoDG Droplet Digital PCR System to partition samples into droplets of roughly 1 nL using default settings. Amplification was performed on a BioRad T100 thermocycler using the following thermocycling program: 50 °C for 60 min, 95 °C for

10 min, 40 cycles of 94 °C for 30 s and 55 °C for 1 min, followed by 1 cycle of 98 °C for 10 min and 4 °C for 30 min with ramp speed of 1.6 °C/s at each step[40].

We thresholded the samples to ascertain the value at which a droplet was considered positive by applying a multistep process that used the following positive and negative controls included on each plate: ATCC SARS-CoV-2 RNA, RNA extracted directly from attenuated BCoV vaccine prepared in PBS, water, and elution buffers. First, we set the threshold between the mean positive and negative amplitudes of these controls to minimize detected copies in the negative controls and to reflect the expected RNA concentration of the positive samples. We then calculated the difference between the mean negative amplitude and the threshold amplitude in the negative control reactions and added it to the mean negative amplitude for each sample. Positive and negative examples of the final thresholded data are included in Supplementary Fig. 1. Further, we noted the highest detected copy number in the none (no RNA or BCoV spiked) stool samples as the LoB. Samples with detected copies per µL below the LoB were marked as Undetermined. Finally, absolute quantification of nucleic acids using ddPCR relies on the generation of a Poisson distribution of template RNA in droplets, requiring an adequate number of droplets with a negative amplification signal. Therefore, in instances where a reaction has saturated amounts of template, we diluted the sample and performed the assay again to ensure reliable quantification. These dilutions are listed where they were performed. Final copy numbers are reported as copies per µL of target in eluate. This was calculated by multiplying by copies per µL reported in each ddPCR reaction by total reaction volume (22 µL) and dividing by input template volume (5.5 µL).

**Step-by-step procedure**. A step-by-step protocol describing the pipeline from sample preparation to quantification and analysis can be found at Protocol Exchange[41].

**Measurement of dry weight**. We first recorded the weight of one 1.5 mL DNA LoBind microcentrifuge tube per sample. Next, we took two biopsy punches using the Integra Miltex Biopsy Punches with Plunger System (Thermo Fisher Scientific; Catalog # 12-460-410) from each of the relevant stool samples and transferred these to microcentrifuge tubes corresponding to the respective samples. The tubes were then weighed, and the respective wet weight was calculated upon subtracting the weight of the empty tube. Next, they were incubated on a heat block at 100 °C for 72 h and reweighed. The dry weight was calculated by subtracting the weight of the empty tube.

**Data analysis and statistics**. We performed statistical analyses using R (version 4.0.0). All statistical analyses were two-sided, performed on the $\log_{10}$-transformed data prior to rounding, and statistical significance was assessed at $\alpha = 0.05$. Unless otherwise stated, we performed the paired $T$ tests in all comparisons. Linear regressions and paired $T$ tests were performed using the stats (version 4.0.0) and rstatix (version 0.7.0) packages using default parameters and commands. No custom code was used. Regressions were plotted using the ggpubr package (version 0.4.0).

**Reporting summary**. Further information on research design is available in the Nature Research Reporting Summary linked to this article.

## Data availability
All data supporting the findings of this study are available within the paper and in the associated files. Source data are provided with this paper.

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

## Acknowledgements

We thank Jason Andrews, Nasa Sinnott-Armstrong, and Renu Verma for guidance on processing stool samples and detection of RNA using RT-qPCR; Angela Rogers and Julie Parsonnet for providing stool and rectal swab samples from patients admitted at Stanford Hospital; Scott Jackson and Stephanie Servetas from NIST for providing us aliquots of standardized stool; the Applied Genetics Group at NIST for aliquots of the SARS-CoV-2 synthetic RNA; Dean Felsher for access to the QuantStudio 12K Flex qPCR machine; Yvonne Maldonado and Jonathan Altamirano for helping acquire funding to support toward this work; Said Attiya and Dhananjay Wagh for guidance on applying ddPCR assays; David Solow-Cordero for assistance setting up the Biomek FX and providing access; Luisa Jiminez and Sopheak Sim for assistance in using the Stanford Functional Genomics Facility and High-Throughput Bioscience Center; Hua Tang for generous guidance in biostatistics; and Ryan Park for assistance in analyzing RT-qPCR data generated as part of the Lambda study. We are grateful to the Peginterferon-λ1a clinical trial team for coordinating procurement of stool samples from outpatients enrolled in this trial. Biorender has been a valuable resource to creating schematic illustrations. This work was supported by a ChemH-IMA grant (to A.S.B. and P.J.) and the Stanford Dean's Postdoctoral Fellowship (to A.N.). A.S.B. laboratory is supported by NIH R01 AI148623 and R01 AI143757. A.H. is supported by an NSF Graduate research fellowship program grant.

## Author contributions

A.N. and A.H. designed and performed experiments, analyzed data, and wrote the manuscript; S.Z. helped design experiments and analyze data; E.F.B. and S.E.V. collected stool samples and helped design experiments and analyze data; M.W. and A.B. guided ddPCR assays and analysis of related data; U.S. and P.J. helped plan and collect patient samples in the Lambda study; B.A.P. guided RNA extraction and setting up of RT-qPCR assays; A.S.B. helped design experiments, analyze data, and wrote the manuscript.

## Competing interests

A.N., A.H., S.Z., E.F.B., S.E.V., M.W., A.B. and A.S.B. are co-inventors on a U.S. provisional patent application #63/172,045 that has been filed and relates to the methods presented in this manuscript. The other authors declare no competing interests.
