## [Peer Review File · Nature Communications]

Reviewers' Comments:

Reviewer #1:

Remarks to the Author:

Dr. Natarajan and colleagues report on experimental data related to the collection/preservation, extraction and amplification of stool samples utilized for molecular testing for SARS-Cov2 RNA by conventional RT-PCR as well as ddPCR. The work is well written, organized and experimentally addresses many currently unknown factors regarding the testing of stool for SARS-COV-2. Their data suggest the need for appropriate preservative matrices and extraction kits and reviews performance characteristics of a few of commonly available kits. One major drawback to the generalizability of this study relates to the use only a small number of actual clinical isolates used from COVID+ patients. Spiked stool does not account to the GI transit which likely plays an important role in what material is actually shed.

Points to consider:

- 1) An important use of molecular testing of stool for SARS-CoV-2 RNA is the monitoring of stool banks of samples to be utilized for fecal microbiota transplantation and is actively being looked at by FDA who oversees these products for patient use.
- 2) With regard to spiked SARS and BCoV in stool using PBS alone (Page 15 PBS panels) there appears to be a marked difference between the non-preservative (PBS) performance between the two targets with SARS-COV-2 heavily impacted and BCoV much less so. Can the authors comment whether the spiked in material itself may be responsible. (ie could poor performance simply be naked SARS-CoV-2 RNA vs Encapsulated BCoV. If this were the rationale then performance with actual virus may not be as hampered)
- 3) Along with point 2 above; were the COVID stool samples from the peginterferon patients able to be tested by PCR in PBS alone without preservative? This would allow for a true assessment of the impact/necessity of the preservative formulations.
- 4) While the graphs and data of the RT-PCR data (page 12 and supplementary data sheet 1) show linearity of the dilution series, a major assessment of PCR reactions, especially as it relates to stool, relates to reaction efficiencies. The reaction efficiencies should be calculated from the slopes and can both be used to assess the presence of inhibitors as well as compare across reaction/sample-prep conditions.

Minor Points:

- 1) It is noted that a replicate of one of the RT-PCR samples at the highest concentration failed to amplify (page 12 panel B) is this correct? Is there an explanation of the failure of this high VL sample to amplify?
- 2) Gaebler's suggestion (Ref 7) that his finding of persistence of both antigen and RNA from SARS-CoV-2 could be indicative of continued antibody evolution is hypothetical and their finding is not truly "evidence that prolonged shedding of SARS-COV-2 in stool may be linked to an improved immune response" (page 3 line 56 and page 29 line 615). In fact GI persistence may be explained as failure of immunological clearance with retained antigen/RNA (their data of its presence was never correlated with patient recovery or immunologic status).

Reviewer #2:

Remarks to the Author:

Strength

1. This is a very comprehensive study on the preservation of fecal viral RNA for RT-PCR detection using synthetic RNA.
2. The paper is well written.

Weakness

1. There is no innovation. The finding is quite expected.
2. The number of samples in the real life evaluation in the field setting is far too low for a proper field evaluation.

Reviewer #3:

Remarks to the Author:

With great pleasure I have reviewed the manuscript entitled "Standardized and optimized preservation, extraction and quantification techniques for detection of fecal SARS-CoV-2 RNA" for Nature Communications.

The authors have developed and validated a clear and optimized blueprint/methodology for SARS-CoV-2 viral RNA detection in stool samples. The availability of a standardized method is of importance for daily practice and research, enabling not only reliable outcomes but also comparison of results.

The analyses and methodology are clear, and the final outcomes are robust and based on exhaustive testing of different (combined) strategies. This led to the nice illustration/recommendation (Figure 7, supplementary data), which is of direct practical use in many laboratory facilities worldwide.

The work is not really novel or innovative, there are other manuscripts available aimed at measuring SARS-CoV-2 RNA in stool samples (for example: A method for detection of SARS-CoV-2 RNA in healthy human stool: a validation study - The Lancet Microbe).

Incorporating or combining this developed method with the (yet unpublished) clinical data from the Peginterferon Lambda-1a trial would increase its newsworthy for a broader group of scientists/readers.

From a more patient-centered view, it would be interesting to describe and validate this technique for (home-based) anal swabs as well.

Lastly, I want to thank the authors for all their hard work in these challenging times.

Reviewer #4:

Remarks to the Author:

In this study, the authors attempt to find a standardized best practice for storing and processing fecal samples for the detection of SARS-CoV-2 RNA. Surveying sewage for SARS-CoV-2 RNA is an important component of public health surveillance and detection. The authors chose multiple commonly used preservatives, RNA extraction kits, and both ddPCR and RT-qPCR to land on a comprehensive methodology to detect SARS-CoV-2 RNA from fecal samples. This is a well written manuscript that addresses a clear knowledge gap and can go a long way in standardizing now commonly used approaches. As the authors point out in the introduction, their approach can lead to data harmonization across studies and allow for a metanalysis. See my comments below.

Abstract: Nothing to add

Introduction:

Line 50- Is there a preprint the authors could link to and/or another study suggesting the long-term detection of SARS-CoV-2 in the stool of previously infected individuals? I don't believe it is best practice to reference unpublished observations.

Methods

Please describe how the SARS-CoV-2 primer/probe assay was designed. Were commonly used targets used? Or were they made de novo by the group and now being published?

Line 159- how often was a CT value detected in one of the various controls listed, and what were the CT values? Ideally no CT value would be detected in the controls.

Results

I agree and appreciate the first part of the results section- positive controls through a standard curve of either synthetic RNA or quantified virus is incredibly important.

Line 238- Could the authors elaborate on failure of 1 in 8 reactions using the NIST RNA standard? Was this run 8 total times and failed once? Or was this run many times and failed on average 1 in 8 reactions?

The abbreviations I found a bit confusing, specifically the ZY and ZV describing both Zymo products. It may be helpful for the reads to see the name of the kits you are referring to in the title sentences in the results section.

Line 301- technically this assay is only designed to measure quantities of SARS-CoV-2 RNA, not total RNA from the stool sample. Were any other end points measured in these experiments (e.g. total RNA quantification via Qubit or a common human target such as RNase P for RT-qCR)?

Line 533-534. Could you please elaborate here? Was this sample not supposed to have SARS-CoV-2 RNA in it, but it was detected in the sample with bovine COV? It is not clear to me how to interpret this result.

Miscellaneous

Although potentially outside the scope of this manuscript- it would be interesting to apply similar standardization assays to wastewater. This manuscript is limited to stool coming from single individuals, but the public health surveillance important of detecting SARS-CoV-2 RNA in fecal matter is at the scale of wastewater. I would encourage the authors to consider doing similar experiments for wastewater surveillance, potentially in a follow-up manuscript.

REVIEWER COMMENTS

Reviewer #1 (Remarks to the Author):

Dr. Natarajan and colleagues report on experimental data related to the collection/preservation, extraction and amplification of stool samples utilized for molecular testing for SARS-Cov2 RNA by conventional RT-PCR as well as ddPCR. The work is well written, organized and experimentally addresses many currently unknown factors regarding the testing of stool for SARS-COV-2. Their data suggest the need for appropriate preservative matrices and extraction kits and reviews performance characteristics of a few of commonly available kits.

We thank the reviewer for noting the importance and comprehensiveness of our work. We are glad that the reviewer agrees with us that the current work addresses many unknown factors regarding testing stool for SARS-CoV-2 RNA.

One major drawback to the generalizability of this study relates to the use only a small number of actual clinical isolates used from COVID+ patients. Spiked stool does not account to the GI transit which likely plays an important role in what material is actually shed.

We fully agree with the reviewer that transit of viral RNA through the GI can lead to effects that are not accounted for in the spiked samples. We have now included this important consideration in the relevant results section as *“Additionally, transit of the viral RNA through the GI tract may affect its detection in a manner not captured by the spiked-in stool samples.”*

We appreciate the concern about the generalizability of this study to clinical samples. We have addressed this concern with additional experiments as summarized here.

a) Testing of extraction kits. We carried out 90 additional extractions and assays, and have now extended the comparison of the extraction kits from 5 to 20 sample sets (revised Figure 5). In total, we have now extracted 120 RNA samples from 20 independent COVID-19 positive patients through three different extraction kits, both with and without BCoV as a spike in control. Reassuringly, the results we see in the larger sample set are concordant with our previous conclusion - the QA extraction kit continues to perform the best, and the addition of BCoV as a spike in control does not impact the accurate detection of SARS-CoV-2 RNA (Revised Fig 5 and revised Supplementary Fig 7).

b) Testing of preservation kits. Since the original manuscript was submitted, we have now collected and processed more clinical stool samples from COVID-19 patients in both the OG and ZY preservatives. Therefore, the revised manuscript now compares the performance of the top two preservatives, OG and ZY, across 188 paired clinical stool samples collected from 90 COVID-19 positive patients (Figure 4). Correspondingly, we have revised the writing of this section as follows - *“Out of 188 such samples from 90 independent COVID-19 outpatients, 94 stool samples did not yield a detectable amount of target RNA in either preservative and were left out of further analysis. Taking the 94 paired samples from which we detected the viral RNA targets in at least one of the preservatives, we plotted their log₁₀-transformed concentrations in*

a scatter plot (Fig. 4). Here we fitted a linear regression, excluding samples that yielded RNA in only one of the two preservatives since these skewed the regression. Notably, 40 of these paired samples yielded detectable RNA only in ZY, in comparison to 7 in only OG.”

Taken together, we are confident that this work is now validated across a sizeable set of clinical samples, and that our conclusions are consistent.

We appreciate the time and effort taken by the reviewer to provide us valuable suggestions.

Points to consider:

1) An important use of molecular testing of stool for SARS-CoV-2 RNA is the monitoring of stool banks of samples to be utilized for fecal microbiota transplantation and is actively being looked at by FDA who oversees these products for patient use.

We thank the reviewer for mentioning this. We fully agree with them and are confident that our work presents a reliable template for monitoring stool banks. Additionally, since this is an important application of the work, we have mentioned this in the discussion section as follows - *“Further, if the limited evidence of possible oral-fecal transmission of SARS-CoV-2 proves true, our ability to reliably test stool samples would be vital to controlling the spread of the pandemic as well as to inform healthcare practices such as fecal microbiota transplants.”*

2) With regard to spiked SARS and BCoV in stool using PBS alone (Page 15 PBS panels) there appears to be a marked difference between the non-preservative (PBS) performance between the two targets with SARS-COV-2 heavily impacted and BCoV much less so. Can the authors comment whether the spiked in material itself may be responsible. (ie could poor performance simply be naked SARS-CoV-2 RNA vs Encapsulated BCoV. If this were the rationale then performance with actual virus may not be as hampered)

We appreciate the close attention paid by the reviewer to this detail, and agree that a comment about this phenomenon is important to the study. Our thoughts on this observation are summarized in the corresponding results section as - *“Across stool spiked with 10^3 copies per μL of sample, we failed to detect any target RNA from PBS-preserved samples. We believe this is because the unpackaged SARS-CoV-2 RNA was degraded by RNAses known to be present in stool. While these data suggest that OG and ZY buffers are critical to preserving naked RNA in stool, testing preservatives in the context of unpackaged SARS-CoV-2 RNA may not be representative of clinical samples. This is because we do not yet know whether SARS-CoV-2 RNA shed in stool is in its naked unpackaged state, protected in an encapsulated structure (such as the virus itself, virus-like particles or host-double membrane vesicles), or a combination thereof.*

“Hence, we sought to identify a proxy for SARS-CoV-2 that is not known to cause disease in humans and is thus safe to handle in the laboratory at biosafety level 1. We picked Bovine coronavirus (BCoV), a virus that belongs to the same genus as SARS-CoV-2, Betacoronavirus, in the subgenus Embecovirus, sharing this taxonomy with other human pathogens (HCoV-

HKU1 and HCoV-OC43). BCoV and SARS-CoV-2 share a common structural architecture, and are both positive stranded RNA viruses.”

Follow-up experiments using attenuated BCoV as a spike-in (Figure 2 panel B) highlight that PBS continues to perform worse than the other preservatives. Notably, the relative performance of PBS to the other preservatives is better when tested with packaged RNA as compared to naked RNA. This further validates our hypothesis and the reviewer’s thoughtful comment.

3) Along with point 2 above; were the COVID stool samples from the peginterferon patients able to be tested by PCR in PBS alone without preservative? This would allow for a true assessment of the impact/necessity of the preservative formulations.

We agree with the reviewer that access to paired stool samples preserved in PBS would make for a perfect comparison. Unfortunately, we were unable to collect fresh stool samples owing to logistical constraints - patients in the Peginterferon lambda study were suffering from mild to moderate COVID-19, were therefore treated as outpatients and requested to collect stool samples at home. Hence, we were unable to have them collect samples without a preservative given storage on ice at home for a variable number of hours before the sample was brought in for aliquoting and freezing would likely compromise the sample. Also, this type of collection (long term storage on ice at home) is much more labor intensive and complicated - and poses greater safety risks to patients. At the early phase of the pandemic when it was rather unclear as to the infectious potential of stool, we strongly preferred to have patients keep only “inactivated” stool samples that had been preserved in a kit at home, as opposed to having raw stool samples, due to potential safety concerns.

4) While the graphs and data of the RT-PCR data (page 12 and supplementary data sheet 1) show linearity of the dilution series, a major assessment of PCR reactions, especially as it relates to stool, relates to reaction efficiencies. The reaction efficiencies should be calculated from the slopes and can both be used to assess the presence of inhibitors as well as compare across reaction/sample-prep conditions.

We thank the reviewer for this very good idea and have now included reaction efficiencies in Supplementary Table 1 and in the relevant parts of the results section as follows:

“The dilution series prepared by two different users working with independent aliquots of the standards revealed ATCC’s synthetic RNA standard to be a reliable control with high concordance across reactions targeting the E, N1 and N2 genes (Fig. 1a,b); efficiencies of associated RT-qPCR reactions are 98 - 101% (Supplementary Table 1). Notably, RdRP proves to be a poor target for the ATCC synthetic SARS-CoV-2 RNA under given reactions conditions, since detection is decreased by an order of magnitude in the ddPCR assay and the RT-qPCR reaction efficiency is compromised (114%). This observation is in keeping with a previous study that found the RdRP primer set to be less sensitive than E, N1 and N2.”

“Prior to stool based testing, we evaluated the performance of the ddPCR and RT-qPCR assays with the recommended primer/probe set to detect the M gene in BCoV RNA. Towards this, we used RNA extracted directly from the attenuated BCoV vaccine prepared in PBS in the absence of stool. We found both the ddPCR and RT-qPCR assays reliably tracked the seven-point ten-fold dilution of the RNA extracts, and the RT-qPCR reaction efficiency of targeting the M gene is 97 - 100% (Supplementary Fig. 4a,b, Supplementary Table 1).”

Minor Points:

1) It is noted that a replicate of one of the RT-PCR samples at the highest concentration failed to amplify (page 12 panel B) is this correct? Is there an explanation of the failure of this high VL sample to amplify?

We appreciate the reviewer for noting this aberration. We believe this is likely due to an experimental error in the RT-qPCR assay and have commented on this as follows - *“Further, one out of a total of eight RT-qPCR reactions assaying the NIST RNA for the E gene at 10⁴ RNA concentration failed to amplify, likely due to an experimental error in the RT-qPCR assay. This result highlights the importance of running RT-qPCR assays in replicates.”*

2) Gaebler’s suggestion (Ref 7) that his finding of persistence of both antigen and RNA from SARS-CoV-2 could be indicative of continued antibody evolution is hypothetical and their finding is not truly “evidence that prolonged shedding of SARS-COV-2 in stool may be linked to an improved immune response” (page 3 line 56 and page 29 line 615). In fact GI persistence may be explained as failure of immunological clearance with retained antigen/RNA (their data of its presence was never correlated with patient recovery or immunologic status).

We agree with the reviewer that Gaebler and colleagues do not present conclusive evidence that prolonged shedding was linked to improved immune response. In response to the reviewer’s keen comments, we have now revised both references to this work to explicitly acknowledge the preliminary nature of this hypothesis. These read as follows -

In the introduction - *“However, from an individual patient health perspective, SARS-CoV-2 antigen is found to persist in the GI tract. Further, there is a preliminary hypothesis that persistent SARS-CoV-2 RNA and protein antigen in the GI tract may promote evolution of host humoral immunity to variants of the virus. Relatedly, prolonged viral RNA shedding in stool may indicate a superior immune response.”*

In the discussion - *“With preliminary emerging evidence that prolonged shedding of SARS-CoV-2 RNA in stool may be linked to an improved immune response, there may be an opportunity to leverage fecal testing of RNA as a prognostic marker.”*

Reviewer #2 (Remarks to the Author):

Strength

1. This is a very comprehensive study on the preservation of fecal viral RNA for RT-PCR detection using synthetic RNA.
2. The paper is well written.

We thank the reviewer for finding our work to be comprehensive on the preservation of fecal viral RNA for RT-PCR detection and for appreciating our presentation of this work.

Weakness

1. There is no innovation. The finding is quite expected.

We appreciate that the reviewer finds this work not to be innovative. Rigorously validated experimental techniques, while not always innovative, certainly form a valuable foundation for sound science. In that regard, we are glad to note that the reviewer finds this work comprehensive, meeting our objective of presenting the most comprehensive evaluation of existing methods for stool preservation, viral RNA extraction and detection.

While we appreciate that the reviewer finds these results to be expected, we believe this to be the benefit of hindsight. Evidence of this is borne out in existing literature - a) majority of studies that have analyzed SARS-CoV-2 viral RNA in stool have saved samples without a preservative - a strategy we find to be ineffective in this current work, b) there is no comparison or benchmarking of existing viral RNA extraction methods, let alone an understanding of how this interacts with samples preserved in different buffers, c) emerging literature in this space, time and again, apply non-standardized data analysis - few studies incorporate controls to establish the limit of blank, no studies compare the relative performance of ddPCR to RT-qPCR, and few studies include internal controls to validate the extraction methodology.

Taken together, we are confident that the current work presents a reliable, externally validated methodological template for future studies of SARS-CoV-2 RNA and other viral RNAs in stool - critically bridging a knowledge gap in the current literature.

2. The number of samples in the real life evaluation in the field setting is far too low for a proper field evaluation.

We appreciate the importance of rigorous field evaluation, and therein, the need for a sizeable sample set. We have addressed this concern with additional experiments as summarized here.

a) Testing of extraction kits. We carried out 90 additional extractions and assays, and have now extended the comparison of the extraction kits from 5 clinical sample sets to 20 clinical sample sets (revised Figure 5). In total, we have now extracted 120 RNA samples from 20 independent COVID-19 positive patients through three different extraction kits, both with and without BCoV as a spike in control. Reassuringly, the results we see in the larger sample set are concordant with our previous conclusion - the QA extraction kit continues to perform the best,

and the addition of BCoV as a spike in control does not impact the accurate detection of SARS-CoV-2 RNA (Revised Fig 5 and revised Supplementary Fig 7).

b) Testing of preservation kits. Since the original manuscript was submitted, we have now collected and processed more clinical stool samples from COVID-19 patients in both the OG and ZY preservatives. Therefore, the revised manuscript now compares the performance of the top two preservatives, OG and ZY, across 188 paired clinical stool samples collected from 90 COVID-19 positive patients (Figure 4). Correspondingly, we have revised the writing of this section as follows - *“Out of 188 such samples from 90 independent COVID-19 outpatients, 94 stool samples did not yield a detectable amount of target RNA in either preservative and were left out of further analysis. Taking the 94 paired samples from which we detected the viral RNA targets in at least one of the preservatives, we plotted their log₁₀-transformed concentrations in a scatter plot (Fig. 4). Here we fitted a linear regression, excluding samples that yielded RNA in only one of the two preservatives since these skewed the regression. Notably, 40 of these paired samples yielded detectable RNA only in ZY, in comparison to 7 in only OG.”*

Taken together, we are confident that this work is now validated across a sizeable set of clinical samples, and that our conclusions are consistent.

Reviewer #3 (Remarks to the Author):

With great pleasure I have reviewed the manuscript entitled "Standardized and optimized preservation, extraction and quantification techniques for detection of fecal SARS-CoV-2 RNA" for Nature Communications.

The authors have developed and validated a clear and optimized blueprint/methodology for SARS-CoV-2 viral RNA detection in stool samples. The availability of a standardized method is of importance for daily practice and research, enabling not only reliable outcomes but also comparison of results.

We thank the reviewer for noting the importance of an optimized method for fecal SARS-CoV-2 RNA extraction and detection. We share an appreciation for the importance of a standardized method both for primary research and to enable cross-study harmonization.

The analyses and methodology are clear, and the final outcomes are robust and based on exhaustive testing of different (combined) strategies. This led to the nice illustration/recommendation (Figure 7, supplementary data), which is of direct practical use in many laboratory facilities worldwide.

We appreciate this thoughtful comment from the reviewer. We are glad they found this work to be comprehensive, and Supplementary Figure 7 to be a valuable resource.

The work is not really novel or innovative, there are other manuscripts available aimed at measuring SARS-CoV-2 RNA in stool samples (for example: A method for detection of SARS-CoV-2 RNA in healthy human stool: a validation study - The Lancet Microbe).

We thank the reviewer for highlighting work from Coryell and colleagues in The Lancet Microbe. We appreciate that this study does advance our knowledge about the influence of stool storage parameters on the detection of SARS-CoV-2 RNA. They test the effectiveness of four different buffers at preserving stool derived from a single donor spiked with SARS-CoV-2 RNA, using the QiaAMP viral extraction kit and RT-qPCR based quantification. Further, they acquire 8 stool and 14 anal swabs, stored without any preservative, to validate conclusions from their preliminary experiments.

Our data here complements work from Coryell and colleagues in notable ways, summarized in the table below. Briefly, we test a combination of three preservation methods, with three extraction kits and two different detection methods - noteworthy is that the preservation and extraction kits interact in different ways affecting the detection of fecal SARS-CoV-2 RNA. In addition, we present a valuable template for establishing the limit of blank, and using BCoV as a control. Further, we extend our analysis to clinical samples, testing preservatives across 172 stool samples from 113 unique COVID-19 patients, and the extraction methods across 20 stool samples from unique patients. Notably, these stool samples are all from adult patients, the primary demographic affected by COVID-19, different from the pediatric cohort used by Coryell *et al.*

We believe that the current work complements that from Coryell *et al* and provides a holistic understanding of the complete pipeline for the storage, extraction and detection of viral RNA from stool samples.

	Coryell et al., 2021.	Natarajan, Han et al., 2021.
Control Samples	One healthy human stool donor	 • Two healthy human stool donors on different diets • Standardized stool from NIST
Clinical stool samples	 • 8 pediatric stool samples • 14 pediatric anal swabs 	 • Testing preservatives - 172 stool samples from 113 unique adult COVID-19 outpatients patients • Testing extraction methods - 20 stool samples from unique adult COVID-19 outpatients
Extraction kits tested	QiaAMP Viral Extraction	 • QiaAMP Viral Extraction • MagMAX Viral Pathogen • Zymo Quick RNA Viral
Preservatives Tested	 • PBS • Roche STAR Buffer • Cary-Blair medium • Zymo DNA/RNA Shield 	 • PBS • OMNIgene GUT • Zymo DNA/RNA Shield
Amplification Methods	RT-qPCR	 • RT-qPCR • ddPCR
Detection/Thresholding	>40 Cq set as cutoff for true positive assay	Derived from a combination of limit of blank, standard curve, and negative control samples
Internal Control	Human RNase P	Bovine Coronavirus Spike-In

Incorporating or combining this developed method with the (yet unpublished) clinical data from the Peginterferon Lambda-1a trial would increase its newsworthy for a broader group of scientists/readers.

We appreciate this suggestion from the reviewer. The method developed here is indeed being used to characterize viral RNA shedding in longitudinal samples from a clinical trial of Peginterferon Lambda-1a. Two reasons we opted not to combine this methods development work with the final results of the Peginterferon Lambda-1a study:

- 1) We are still in the process of collecting long-term longitudinal stool samples from participants of this clinical trial. Since, we believe that methodological insight from this current work is

immediately valuable, we did not want to delay this manuscript until the sample collection effort was completed.

- 2) We worry that the many nuances in this extensive methods development effort and knowledge thereof may become lost in the supplement of a largely clinical manuscript oriented to a clinical audience.

From a more patient-centered view, it would be interesting to describe and validate this technique for (home-based) anal swabs as well.

We agree that this would be interesting to investigate. Unfortunately, none of the many ongoing COVID-19 related studies at Stanford employ a collection of home-based anal swab samples. We hope that our proposed pipeline can be investigated for its generalizability to home-based anal swabs in the future through studies that involve anal swab collection.

Lastly, I want to thank the authors for all their hard work in these challenging times.

Thank you so much for taking the time and effort to provide us valuable suggestions. We are grateful for your input.

Nanne de Boer
Department of Gastroenterology and Hepatology
Amsterdam UMC, AGEM Research Institute
Amsterdam, The Netherlands

Reviewer #4 (Remarks to the Author):

In this study, the authors attempt to find a standardized best practice for storing and processing fecal samples for the detection of SARS-CoV-2 RNA. Surveying sewage for SARS-CoV-2 RNA is an important component of public health surveillance and detection. The authors chose multiple commonly used preservatives, RNA extraction kits, and both ddPCR and RT-qPCR to land on a comprehensive methodology to detect SARS-CoV-2 RNA from fecal samples. This is a well written manuscript that addresses a clear knowledge gap and can go a long way in standardizing now commonly used approaches. As the authors point out in the introduction, their approach can lead to data harmonization across studies and allow for a metanalysis. See my comments below.

We thank the reviewer for noting the importance of an optimized method for preservation, extraction and quantification of fecal SARS-CoV-2 RNA. We appreciate their note about the comprehensiveness of the current work and its value to the community.

Abstract: Nothing to add

Introduction:

Line 50- Is there a preprint the authors could link to and/or another study suggesting the long-term detection of SARS-CoV-2 in the stool of previously infected individuals? I don't believe it is best practice to reference unpublished observations.

Thank you for this suggestion. We have now revised the manuscript as follows, citing an existing study - *"Further, patients shed viral RNA in their stool up to 70 days after disease onset, well after they have cleared the infection from their respiratory tissues⁵"*

Methods

Please describe how the SARS-CoV-2 primer/probe assay was designed. Were commonly used targets used? Or were they made de novo by the group and now being published?

We thank the reviewer for pointing out this oversight. We used primers and probes that had previously been published and validated across multiple studies. We have now included appropriate citations in Table 1, which lists all the primers and probes.

Line 159- how often was a CT value detected in one of the various controls listed, and what were the CT values? Ideally no CT value would be detected in the controls.

We appreciate the reviewer's attention to detail here. Our experimental design included a full five-point ten-fold dilution of the SARS-CoV-2 synthetic standard from ATCC as a standard curve on every plate to both establish a limit of detection and quantify viral load.

In the experiment that used standardized stool from NIST, RT-qPCR detected a Cq value for the N1 target in 6 of the 18 unspiked stool samples. The lowest of these was 35.118, which fell above the Cq value detected in our standard curve i.e this Cq value was larger than that

detected for the control sample with 1 copy of RNA per μL ($C_q = 32.831$). No BCoV M gene was detected in the unspiked samples. Complete data towards these are listed in Supplementary Information 1, Sheet 5.

In the experiment that used non-standardized healthy stool, RT-qPCR detected a C_q value for the N1 target in 4 of the 36 unspiked stool samples. The lowest of these was 35.604, which also fell above the C_q value detected in our standard curve i.e this C_q value was larger than that detected for the control sample with 1 copy of RNA per μL ($C_q = 33.875$). Again, no BCoV M gene was detected in the unspiked samples. Complete data towards these are listed in Supplementary Information 1, Sheet 7.

We find that it is not uncommon for negative samples to provide a high C_q value in RT-qPCR experiments. This is why having a robust set of controls to reliably establish a limit of blank is important. Especially in light of the fact that many studies looking at fecal shedding of SARS-CoV-2 RNA do not conform to such rigor, we truly appreciate the reviewer's comment.

Results

I agree and appreciate the first part of the results section- positive controls through a standard curve of either synthetic RNA or quantified virus is incredibly important.

We appreciate that the reviewer shares our perspective regarding the value of a reliable standard for this work. Notably, since previous literature doesn't provide guidance on the reliability of controls, we believe our work will be valuable to the field.

Line 238- Could the authors elaborate on failure of 1 in 8 reactions using the NIST RNA standard? Was this run 8 total times and failed once? Or was this run many times and failed on average 1 in 8 reactions?

We thank the reviewer for highlighting the lack of clarity in this statement. The assay was carried out a total of eight times, out of which it failed once, likely due to an error in the RT-qPCR assay. In order to add clarity, we have now revised the corresponding statement in the manuscript as follows - "*Further, one out of a total of eight RT-qPCR reactions assaying the NIST RNA for the E gene at 10^4 RNA concentration failed to amplify, likely due to an experimental error in the RT-qPCR assay. This result highlights the importance of running RT-qPCR assays in replicates.*"

The abbreviations I found a bit confusing, specifically the ZY and ZV describing both Zymo products. It may be helpful for the reads to see the name of the kits you are referring to in the title sentences in the results section.

We appreciate that it can be hard to keep track of the abbreviations and are thankful for this helpful suggestion. We have now incorporated this change to all the title sentences in the manuscript.

Line 301- technically this assay is only designed to measure quantities of SARS-CoV-2 RNA, not total RNA from the stool sample. Were any other end points measured in these experiments (e.g. total RNA quantification via Qubit or a common human target such as RNase P for RT-qCR)?

We thank the reviewer for noting this. We recognize that we were not clear in the original wording of this sentence. This assay is only designed to quantify SARS-CoV-2 RNA. Therefore, we have now revised this sentence as follows - *“Among the stool preservatives, more SARS-CoV-2 RNA was detected in ZY than OG in both samples spiked...”*

Line 533-534. Could you please elaborate here? Was this sample not supposed to have SARS-CoV-2 RNA in it, but it was detected in the sample with bovine COV? It is not clear to me how to interpret this result.

We thank the reviewer for seeking clarification regarding this observation where we detected SARS-CoV-RNA in the BCoV spiked sample but not in the non-spiked sample. The original stool sample used herein was derived from a COVID-19 positive patient the day after their initial diagnosis. At this stage, patients often shed viral RNA in their stool. In preparing samples for this study, one fraction of the original stool sample was spiked with BCoV while another was not. Then, RNA was extracted using the MM kit from both preparations and assayed using ddPCR. It is unclear to us why the two sample preparations behaved differently. We now elaborate on this point in the manuscript, as follows - *“Comparing the yield of SARS-CoV-2 RNA in samples with (Fig. 5a) and without (Supplementary Fig. 7b) spiked in BCoV reveals that the addition of this control does not significantly affect the yield of RNA (Fig. 5b) with the exception of sample # 10 extracted using MM. In this case, where we expected consistency across samples with and without the BCoV spike in, inexplicably, we find that the stool sample with spiked BCoV does not yield SARS-CoV-2 RNA while the unspiked sample does. We reran the ddPCR assay and observed the same result. Notably, the BCoV spiked sample that does not yield detectable SARS-CoV-2 RNA yields BCoV RNA, revealing that the extraction and detection steps are successful. Therefore, we suspect this anomaly observed in one out of the 20 samples tested to be the result of experimental error in sample preparation.”*

Miscellaneous

Although potentially outside the scope of this manuscript- it would be interesting to apply similar standardization assays to wastewater. This manuscript is limited to stool coming from single individuals, but the public health surveillance important of detecting SARS-CoV-2 RNA in fecal matter is at the scale of wastewater. I would encourage the authors to consider doing similar experiments for wastewater surveillance, potentially in a follow-up manuscript.

We appreciate this suggestion from the reviewer. In fact, our collaborators (and co-authors on this work), Alexandria Boehm and Marlene Wolfe, are working on this in a parallel effort.

Thank you so much for taking the time and effort to provide us with these valuable suggestions. We are grateful for your input.

Reviewed by:
Joseph Fauver, Ph.D.
Associate Research Scientist
Yale School of Public Health

Reviewers' Comments:

Reviewer #1:

Remarks to the Author:

I thank Dr. Natarajan and colleagues for their revisions based on the reviewer suggestions. The inclusion of additional clinical material helps to shore up the applicability of the study. The major contribution of the work remains the evidence of differential kit capabilities regarding laboratory considerations of collection, storage and extraction of stool samples. While the identification of superior and inferior kits remains valuable some issue might be taken with the title's assumption that this paper definitively establishes "standardized and optimized" techniques, as there will most certainly remain tremendous variability in collection devices, techniques, workflows, protocols and molecular detection platforms. Perhaps the title can identify the nature of the paper as demonstrating differential performance characteristics across several commercial preservation and extraction manufacturers.

Reviewer #2:

Remarks to the Author:

N.A.

Reviewer #3:

Remarks to the Author:

Again with great pleasure I have read the revised manuscript by Natarajan concerning SARS-CoV-2 RNA detection in fecal samples.

The authors have revised/changed their manuscript, based on the remarks of all reviewers, in a satisfactory way. Especially the (additional) testing in a larger set of samples with unchanged outcomes is reassuring.

My earlier remark on combining these findings with the results of the Peginterferon Lambda-1a trial still stands, as this would surely lead to an increased newsworthiness. The answer by the authors is nevertheless also true as combining these datasets would also lead to a less detailed description of the studied technique.

I have no more remarks or questions.

Reviewer #4:

Remarks to the Author:

The authors addressed all relevant points of my review.

REVIEWER COMMENTS

Reviewer #1 (Remarks to the Author):

I thank Dr. Natarajan and colleagues for their revisions based on the reviewer suggestions.

We thank the reviewer for their time and effort in providing valuable input that has improved our manuscript.

The inclusion of additional clinical material helps to shore up the applicability of the study. The major contribution of the work remains the evidence of differential kit capabilities regarding laboratory considerations of collection, storage and extraction of stool samples.

We thank the reviewer for noting that the additional clinical samples have bolstered the applicability of the study. We are also glad that the reviewer continues to find our testing of the capabilities of different kits for collection, storage and extraction of RNA from stool samples the major contribution of work.

While the identification of superior and inferior kits remains valuable some issue might be taken with the title's assumption that this paper definitively establishes "standardized and optimized" techniques, as there will most certainly remain tremendous variability in collection devices, techniques, workflows, protocols and molecular detection platforms. Perhaps the title can identify the nature of the paper as demonstrating differential performance characteristics across several commercial preservation and extraction manufacturers.

We thank the reviewer for this nuanced input regarding the title of the manuscript. Given this feedback, we took a fresh look at the results of this manuscript and believe that it provides a standardized method for collection, preservation and extraction of viral RNA from stool samples, specifically to enable consistency across studies. However, we agree with the reviewer about their reservations with the usage of the term 'optimized'. Therefore, we have now updated our title to read - "Standardized preservation, extraction and quantification techniques for detection of fecal SARS-CoV-2 RNA".

Reviewer #2 (Remarks to the Author):
N.A.

Reviewer #3 (Remarks to the Author):

Again with great pleasure I have read the revised manuscript by Natarajan concerning SARS-CoV-2 RNA detection in fecal samples.

We thank the reviewer for their time and effort in providing valuable input that has improved our manuscript.

The authors have revised/changed their manuscript, based on the remarks of all reviewers, in a satisfactory way. Especially the (additional) testing in a larger set of samples with unchanged outcomes is reassuring.

We are glad that the reviewer finds that we have addressed all the remarks from reviewers in a satisfactory manner. We agree that it is reassuring that the additional sample set validates our previous findings from a smaller set of samples.

My earlier remark on combining these findings with the results of the Peginterferon Lambda-1a trial still stands, as this would surely lead to an increased newsworthiness. The answer by the authors is nevertheless also true as combining these datasets would also lead to a less detailed description of the studied technique.

We thank the reviewer for their thoughtful comment here and are glad that they find our response to be satisfactory.

I have no more remarks or questions.

Reviewer #4 (Remarks to the Author):

The authors addressed all relevant points of my review.

We thank the reviewer for their time and effort in providing valuable input that has improved our manuscript.